# Large protein databases reveal structural complementarity and functional locality

Paweł Szczerbiak [1,2], Lukasz M. Szydlowski[1,2], Witold Wydmański[2,3], P. Douglas Renfrew[4], Julia Koehler Leman [5] & Tomasz Kosciolek [1] ✉

Recent breakthroughs in protein structure prediction have led to a surge in high-quality 3D models, highlighting the need for efficient computational solutions. In our work, we examine the structural clusters from the AlphaFold Protein Structure Database (AFDB), a high-quality subset of ESMAtlas, and the Microbiome Immunity Project (MIP). We create a single cohesive low-dimensional representation of the resulting protein space. We show that, while each database occupies distinct regions, they collectively exhibit significant overlap in their functional profiles. High-level biological functions tend to cluster in particular regions, revealing a shared functional landscape despite the diverse sources of data. By creating a representation of protein structure space, localizing functional annotations within this space, and providing an open-access web-server for exploration, this work offers insights for future research concerning protein sequence-structure-function relationships, enabling biological questions to be asked about taxonomic assignments, environmental factors, or functional specificity. This approach is generalizable, thus enabling further discovery beyond findings presented here.

The tectonic shift that has occurred in structural and molecular biology thanks to the abundance of high-quality protein 3D structure models calls for a new outlook on protein structures as a source of information about biology and a tool to understand it further. Thanks to decades of work on protein sequences in the field, efficient methods to store, search, cluster or annotate sequences exist and are well-documented[1–6]. Since the release of AlphaFold2[7] and accompanying AlphaFold Protein Structure Database (AFDB), the number of readily available protein structures grew 1000-fold from ~200k to ~200 M[8,9]. These data found its way into UniProt and other repositories, where they are accessible through text-based or web-based interfaces. Still, just as protein sequence information became ubiquitous and purpose-made methods to leverage this source of biological data became necessary, there is a need for purpose-built tools and a better understanding of the protein structure space. So far, we have seen a rapid development of efficient tools for protein structure processing, e.g. clustering[10], compression[11,12] or representation[13–15]. Subsequently, the

ESMAtlas repository[16], featuring over 600 million predictions, further expanded the available structural models, bringing the total, including those from the Protein Data Bank[17], to nearly 1 billion structures or models. Much effort has been put so far into building a deeper understanding of the AFDB structure space ranging from novel fold search[18,19], domain identification[20], clustering[21], and structure space exploration[22]. But, we still lack a comprehensive and complementary understanding of the protein structure space and how different sources of structure data – obtained using different methods, each utilizing various underlying sequence databases – contribute to this space.

Specifically, what's missing is a shared structural and functional space – a unified reference frame that allows proteins from disparate datasets to be compared meaningfully. Such a framework would support cross-dataset biological inference, enabling systematic comparisons across structure predictors, taxonomic groups, and sequence contexts. It would also help in identifying complementarity and

[1]Sano Centre for Computational Medicine, Kraków, Poland. [2]Małopolska Centre of Biotechnology, Jagiellonian University, Kraków, Poland. [3]Faculty of Mathematics and Computer Science, Jagiellonian University, Kraków, Poland. [4]Center for Computational Biology, Flatiron Institute, New York, NY, USA. [5]Open Molecular Software Foundation, Davis, CA, USA. ✉e-mail: t.kosciolek@sanoscience.org

redundancy, revealing which datasets contribute novel folds and which primarily reinforce known structural motifs. By incorporating structure-based function annotations into this space, we can identify where proteins with similar biological roles cluster together, even in the absence of sequence similarity. This makes this space a powerful discovery tool – one that facilitates the exploration of uncharacterized folds, improves functional annotation pipelines, and helps prioritize proteins of interest in applications ranging from metagenomics to protein engineering. Further, this reference frame is extensible, offering a mechanism to incorporate future datasets for direct structural and functional comparison.

In this study, we analyze three large protein structure model datasets originating from distinct sources. The AlphaFold Protein Structure Database (AFDB)[9], based on a sizable part of the UniProt[23], includes protein models from a wide range of organisms, with a significant representation of eukaryotes. The ESMAtlas[16], derived from MGnify[24], focuses exclusively on proteins from metagenomic studies, and therefore predominantly contains prokaryotic data. The Microbiome Immunity Project (MIP) database[25], based on The Genomic Encyclopedia of Bacteria and Archaea (GEBA)[26], primarily consists of short, single-domain proteins ranging from 40 to 200 residues, representing bacterial genomes, and may thus have similarities with ESMAtlas. In contrast, both the AFDB and ESMAtlas datasets include single- and multi-domain proteins, distinguishing them from the MIP dataset. This diversity allows us to explore various regions of the structural space more thoroughly, such as the localization of high- and low-quality (including intrinsically disordered proteins, IDP) predictions or single- and multi-domain proteins (see Datasets paragraph in Methods section for further discussion). The representative sequences and structures are then projected onto a lower-dimensional space according to their structural features and functionally annotated using an updated version of a structure-based function prediction method deepFRI[27]. As such, our study goes well beyond prior attempts to characterize the protein universe[28–32]. Unlike many previous studies, here we focus on full-chain structures, showing significant novelty (i.e., previously unobserved protein structures or folds) coming from multi-domain proteins, and, previously discarded or unavailable, metagenomic sequences primarily represented in ESMAtlas.

In this work, we present a comprehensive analysis of the current protein structure space, integrating data from the most extensive sources available. We show that the protein structure databases are complementary and exhaustive in terms of folds and biological function. We examine the unique characteristics of each database, analyze the distribution of protein functions within this space, and introduce a web server that allows for customized exploration of these structural datasets [https://protein-structure-landscape.sano.science]. This work aims to provide a deeper understanding and contextualization of the current protein structure landscape, enabling further insights into protein function and evolution.

## Results

### Datasets occupy complementary regions in the structure space
Our workflow is outlined in Fig. 1a. To create a comprehensive dataset of protein structures we focused on non-redundant sequences from the major protein structure databases: AFDB50, highquality_clust30 (a high-quality, representative subset of ESMAtlas), and MIP. In the first stage, we eliminated structural redundancy with Foldseek, optimizing parameters for each database individually (see section Structural clustering in the Supplementary Information). For AFDB, we relied on the results from ref. 21, which categorized the structural models into light (mapping onto Pfam) and dark clusters (without hits to Pfam). These clusters include both low- and high-quality models, whereas the highquality_clust30 dataset from ESMAtlas contains only high-quality predictions. Subsequently, all representative structures (excluding AFDB and ESMAtlas singletons) were gathered into one set and

clustered again with Foldseek to remove structural redundancy between databases (see Supplementary Figs. 1–18 and Supplementary Table 1 for details as well as Supplementary Table 2 for a summary). Next, we functionally annotate the dataset using deepFRI and generate Geometricus representations to explore structural features[13,27]. deepFRI has proven to be a reliable method for generating functional annotations, while Geometricus embeds protein structures into fixed-length shape-mer vectors that can be successfully used in downstream tasks[33–35]. Shape-mer features are then reduced to a two-dimensional structure space using PaCMAP[36]. To assess cluster heterogeneity, we define heterogeneous clusters as those containing models from at least two distinct databases. Finally, all information is integrated into a web server.

Figure 1b highlights the key features of the structure space, with all structural representatives (clusters and singletons) as inputs. For more details on the construction, refer to Supplementary Figs. 19–25. We also embedded CATH superfamilies[37] into this space to better understand its organization. Because our dataset includes all proteins from AFDB and ESMAtlas (with only MIP containing mostly single-domain structures), the CATH superfamilies only partially cover this space. Despite this, a clear separation between CATH classes is evident, with the highest density in the alpha/beta region. Except for two small areas in the structure space, we did not observe a gradient in structure length. Additionally, we found a strong correlation between structure length and mean pLDDT (as shown in Fig. 1b, AFDB pLDDT panel; see also Supplementary Fig. 26 for dispersion measures).

Figure 1c compares the fractions of structures from different databases. For a fair comparison with ESMAtlas, we included only high-quality AFDB models. We observed clear complementarity between AFDB, ESMAtlas, and MIP clusters in their coverage of the structure space (we call this structural complementarity). The separation between structures from different databases cannot be explained by structure length alone, as most AFDB and ESMAtlas structures are of similar size (see Supplementary Fig. 17). Interestingly, ESMAtlas and AFDB light proteins largely occupy the same region of the structure space (see Supplementary Fig. 27 for absolute values and Supplementary Fig. 28 for all AFDB models), suggesting that both databases extensively cover the known structural landscape. Additionally, we see significant overlap between AFDB light and dark clusters.

### Protein conformations display gradual shifts across the structural landscape
We selected several representative models from different regions of our dataset to illustrate the structural landscape (Fig. 2). Selected models primarily have high mean pLDDT scores (except for a few, such as A0A498NE77 with a mean pLDDT of 69, and five proteins marked in blue, coming from clusters with long representative structures).

The spatial distribution of the structures highlighted in red indicates the presence of two primary branches that emerge from a central area, predominantly composed of alpha-beta structures. These branches extend into beta-only structures on the left side of the figure and alpha-only structures, which are mainly located on the right and lower-right sections. The long representative structures, shown in blue and primarily consisting of low-quality AFDB models that may include IDPs, are located in the bottom-left corner, where significant divergence and continuity are also observed. However, their in silico characterization poses difficulties for structure-based annotation tools due to the lack of such information regarding IDPs. Overall, the structural landscape exhibits a high degree of coherence, with gradual and incremental variations in structural motifs throughout the space, although some outliers are present.

A particularly interesting region is located in the top left (green dots), characterized by fibril proteins with varying cross-sections. These include square (MGYP000417476362, A0A3G3K1M9), flat (A0A0F6LHQ3, A0A498NE77), irregular (MGYP001600570585,

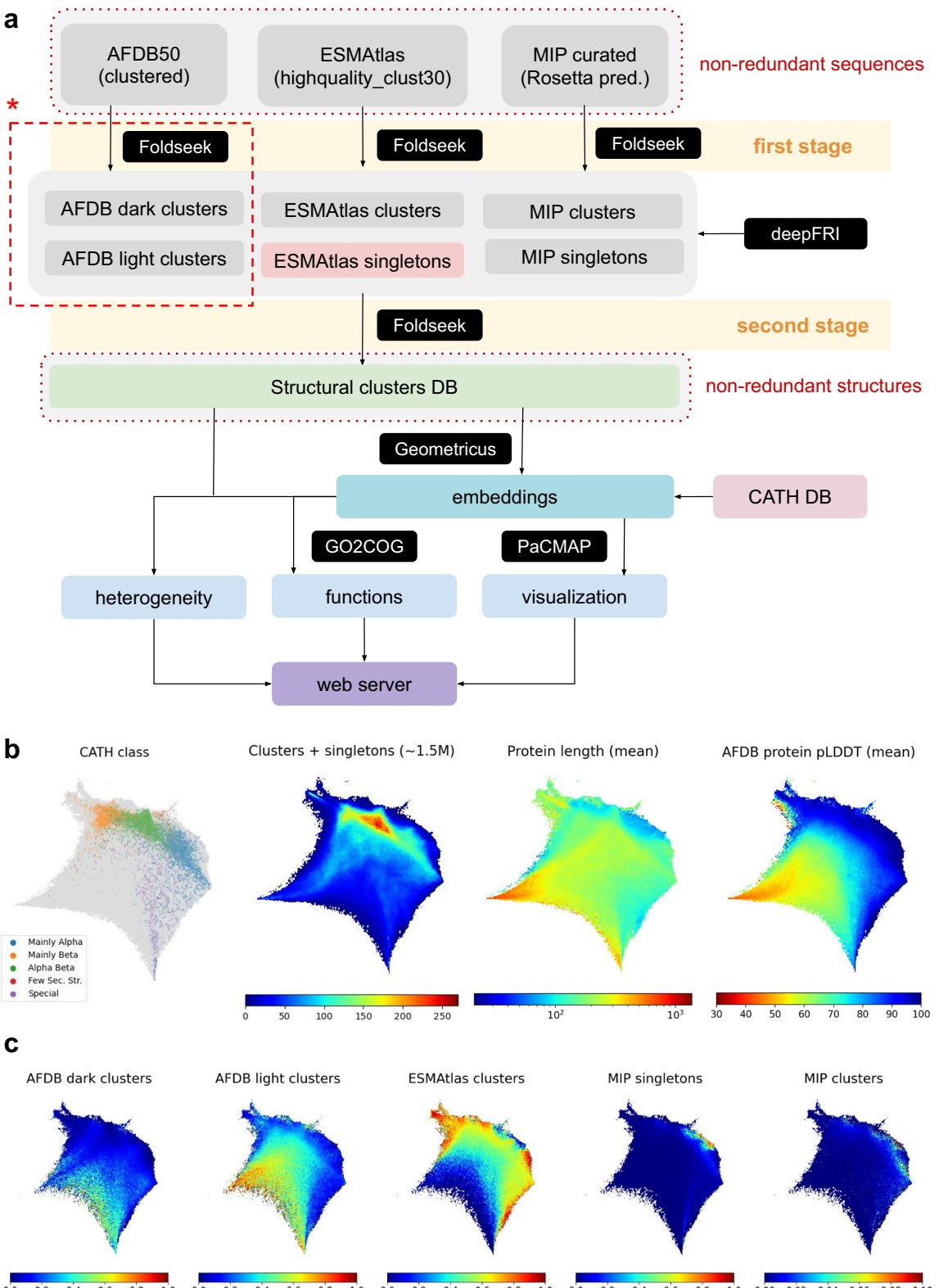

**Fig. 1 | Pipeline and structure space. a** The diagram illustrates the pipeline from top to bottom. We begin from three sets of proteins that are non-redundant at the sequence level and aim to cluster them at the structure level. Red asterisk indicates the part that has been done in ref. 21. Using a similar methodology, we clustered the remaining two databases and, in the second stage, we clustered the cluster representatives (green box) to eliminate structural redundancy between databases. All structures have been annotated with deepFRI and Geometricus to elucidate its functional potential and visualize the structure space, respectively. Dimensional reduction into 2 dimensions has been performed with PaCMAP. Subsequently, we analyzed cluster heterogeneity. **b** CATH class (left), number of structures (middle left), protein length (middle right), and pLDDT for AFDB structures (right) distributed across the structure space (we considered all ~1.5 million cluster representatives + singletons). **c** Fraction of structures coming from a given database (we included only high-quality models i.e. ~3 million structures – see Supplementary Table 2). Panels (**b**) and (**c**) generated using unnormalized Geometricus representations may be found in the Supplementary Information (see Supplementary Figs. 29–32).

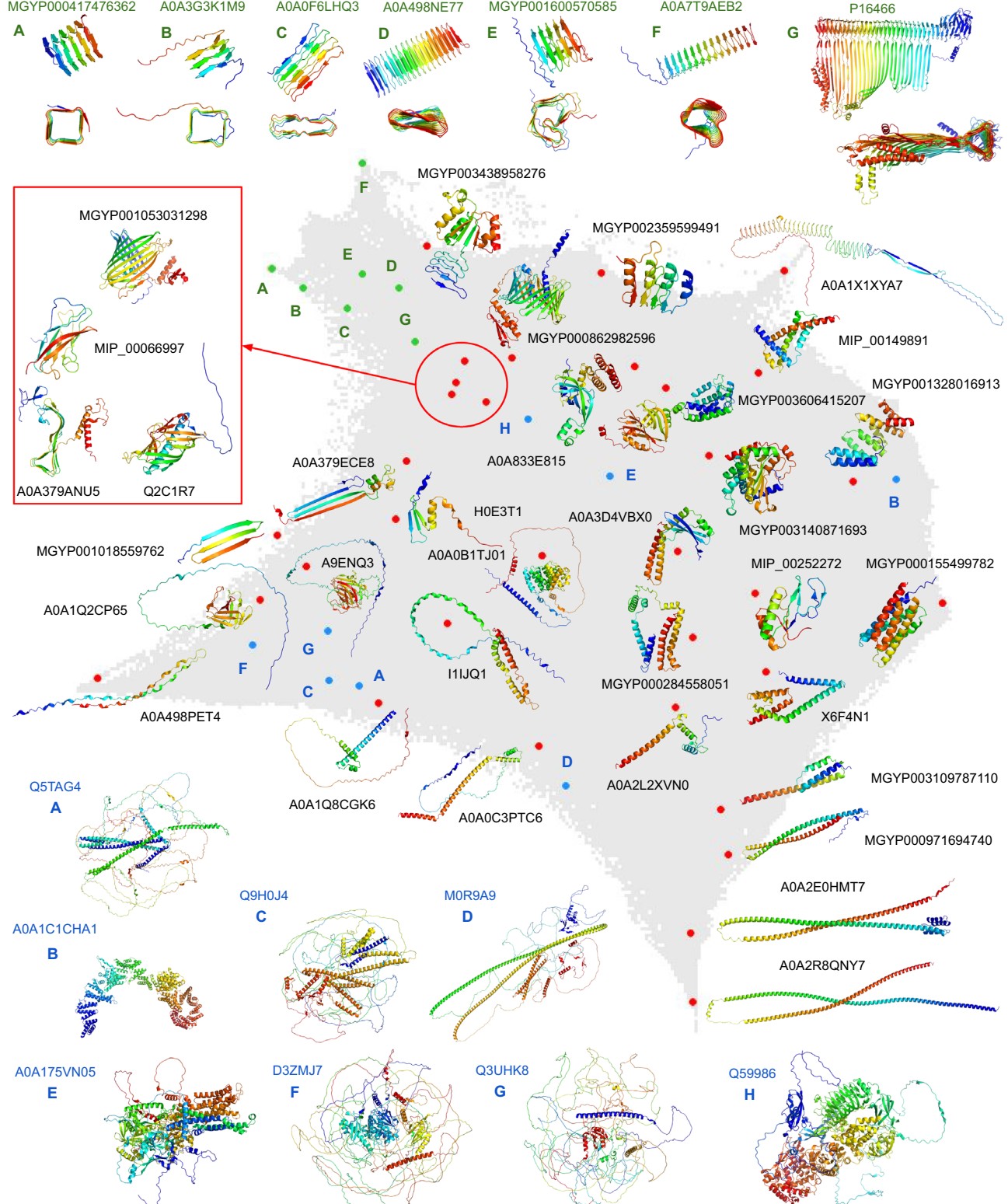

**Fig. 2 | Structural landscape.** Example structures within the protein structure space are represented by red, green, and blue dots. Red dots illustrate structures that are distributed relatively evenly across the space. Green dots, located in the upper left corner with corresponding structures shown from two different perspectives at the top, represent fibril proteins with various cross-sections; these examples were selected with high predicted alignment error to ensure confidence in their overall geometry. Blue dots, with structures displayed at the bottom, represent clusters with long representative structures (see also Fig. 5e, where structures are colored by pLDDT). For easier navigation, blue and green dots are labeled with letters.

A0A7T9AEB2), and even triangular (P16466) cross-sections, which may be responsible for diverse functional roles. The proteins clustered in this region, often involved in pore-forming activity and transmembrane transport, are of particular medical interest due to their association with pathogenicity and virulence, such as hemolysin (P16466). Other fibrillar proteins in this area are associated with translation (A0A3G3K1M9), signaling and defense responses (A0A0F6LHQ3), macromolecule interactions (A0A498NE77), and kinase activity in response to stimuli (A0A7T9AEB2).

To evaluate the generalizability of our dimensionality reduction approach, we applied it to two additional datasets: a smaller set of 10,000 AlphaFold2 models of artificial sequences generated with ProtGPT2[38], and a larger set of 351,242 structural models from the Big Fantastic Viral Database (BFVD)[39]. Results are detailed in the Supplementary Information, subsection Other databases. As shown in Supplementary Figs. 33, 34, ProtGPT2 structures are distributed fairly evenly across the structural landscape, while viral structures – though spanning the entire space – tend to exhibit a higher alpha-helical content. Notably, our method successfully identified structural outliers, such as short unfolded motifs (right panel, Supplementary Fig. 34), which are absent in the curated dataset that includes AFDB, ESMAtlas, and MIP cluster representatives. Interestingly, a substantial portion of both ProtGPT2 and BFVD structures appear as singletons relative to these reference databases (Supplementary Table 4), but due to their low pLDDT scores (Supplementary Figs. 35, 36). These findings indicate that our approach generalizes well and is applicable to other protein structure datasets.

## High-level functional categories are localized in specific regions of the structure space

We performed functional analysis using deepFRI versions 1.0 and 1.1 which predicts Gene Ontology terms along with corresponding confidence scores (see Supplementary Fig. 38)[27]. While both versions share the same architecture, version 1.1 was trained on AFDB structures. This results in a higher number of predictions compared to version 1.0 but also increases the likelihood of false positives (see Functional annotations/deepFRI v1.1 subsection in the Supplementary Information). To map Gene Ontology (GO) predictions from deepFRI onto Clusters of Orthologous Groups (COG), we employed a customized GO-to-COG mapping (see go2cog mapping paragraph in Methods section for details)[27,40,41]. We further combined COGs into three main functional groups: "Cellular processes and signaling" (COGs: D,M,N,O,U,W,X,Y,Z), "Metabolism" (COGs: C,E,F,G,H,I,P,Q) and "Information storage and processing" (COGs: A,B,J,K,L). Functions that were too generic and did not fall into any other category, were classified as "General Function" (COG: R). Validation on the *E. coli* proteome helped establish optimal deepFRI score thresholds (Supplementary Figs. 39, 40).

The combined functional groups, hereby termed as superCOGs (superCOG 1: "Cellular processing and signaling", superCOG 2: "Metabolism", superCOG 3: "Information storage and processing") had functionally annotated proteins from five datasets from Fig. 1c and MIP novel folds as a subset of the entire MIP dataset[25] into superCOG categories, which reveals similar superCOG distributions across the datasets, including smaller sets like MIP novel folds (Fig. 3a). Despite representing previously unseen folds, the MIP novel folds dataset does not consist of unknown functions, as most of these proteins are annotatable. Similar observations were made in AFDB dark clusters, where 67-80% of proteins were annotated. SuperCOG 3 functions (related to information storage and processing, including replication and gene expression) are less abundant in these datasets compared to others. This is expected since these processes are well-conserved and studied across living organisms[42,43].

In Fig. 3b, we present the percentage of proteins annotated with deepFRI v1.0 for various SuperCOG functional categories. The data reveal that proteins belonging to distinct high-level functional groups are concentrated in specific regions of the structural space, a phenomenon we refer to as functional locality. This non-random distribution of normalized functional categories is supported by Moran's I statistic – a measure of spatial autocorrelation ranging from 0 (no autocorrelation) to 1 (strong autocorrelation). Our analysis yielded a Moran's I value indicative of significant spatial clustering, with a *p*-value near zero, confirming that functionally related categories are not randomly distributed but instead exhibit pronounced spatial autocorrelation (see Fig. 3b and Spatial autocorrelation testing paragraph in Methods for details).

DeepFRI v1.1 achieves 100% annotation coverage of all proteins, but due to being trained on the Uniprot dataset, it predicts superCOG 3 functions with less accuracy than deepFRI v1.0, which was trained on Swiss-Prot. In contrast, superCOG 2 and general function categories show strong agreement between versions 1.0 and 1.1 (compare Fig. 3b and S36). We can notice complementarity in superCOG 1 + 3 annotation whereas regions not annotated by v1.0 are mostly predicted by v1.1 as superCOG 1 + 2. Regions representing the general function category, often associated with low-quality predictions or disordered proteins (compare Fig. 1b), are located in the same corner, suggesting that generally this cluster is hard to predict, regardless of the annotation tool. Overlaying functions with CATH classes, we observed that the alpha/beta CATH class primarily overlaps with metabolic functions (superCOG 2). This is consistent with the higher structural complexity of enzymatic proteins, which are often determined by their substrates[44,45]. The largest COG categories for each dataset are shown in Supplementary Fig. 48.

## Exploring functional diversity in novel protein folds

In gene ontology (GO) annotation database, the functional hierarchy is structured as an ontology tree, where Shannon Information Content (IC) serves as a proxy for measuring the specificity of annotations[46] (see Function prediction paragraph in Methods section for more details). In our analysis of MIP novel fold clusters, we identified several proteins with annotations characterized by high IC values, indicating they are associated with more specific GO terms. A higher IC reflects greater information entropy within the GO database, meaning these proteins are annotated with terms that offer more detailed functional insight (see also ref. 33). This suggests that novel folds can still correspond to known functions (Fig. 4). We hereby selected three examples of such proteins that – despite high IC – do not have structural homologs: proteins with tetrahydromethanopterin S-methyltransferase activity (GO:0030269), intramembrane lipid transporter activity (GO:0140303), and pectinesterase inhibitor activity (GO:0046910). The specificity of these GO terms suggests new structural alternatives for the variety of functions, including core metabolism (GO:0030269 and GO:00140303) and cellular processing (GO:0046910). The output of annotation from all core ontologies (BP, CC and MF), generated by deepFRI agree with the predicted function, by e.g. determining its subcellular location, which can be cross-validated e.g. with TMHMM[47]. Moreover, applying knowledge-based information, such as the habitat of the host organism, i.e. to confirm whether this could be a methanogen, as in the case of Fig. 4a or displaying host-pathogen interactions, as in the case of Fig. 4c, helped us verify whether the predicted functions are likely to be present.

Tetrahydromethanopterin S-methyltransferase (Fig. 4a) forms catalytic complexes with cobalamin (vitamin B12) and transfers methyl groups from tetrahydromethanopterin (H4MPT) to coenzyme M. The gene encoding the MtrA subunit of this complex, mtrA, exhibits significant phylogenetic diversity[48,49]. The ability of these proteins to bind complex substrates, such as H4MPT, and form complexes with other proteins likely drives their structural diversity[48–51]. The protein we identified was derived from the genome of *Blautia* sp., a known methanogen, which strengthens our confidence in this prediction. A Foldseek search revealed the highest similarity to another *Blautia* protein in the AFDB

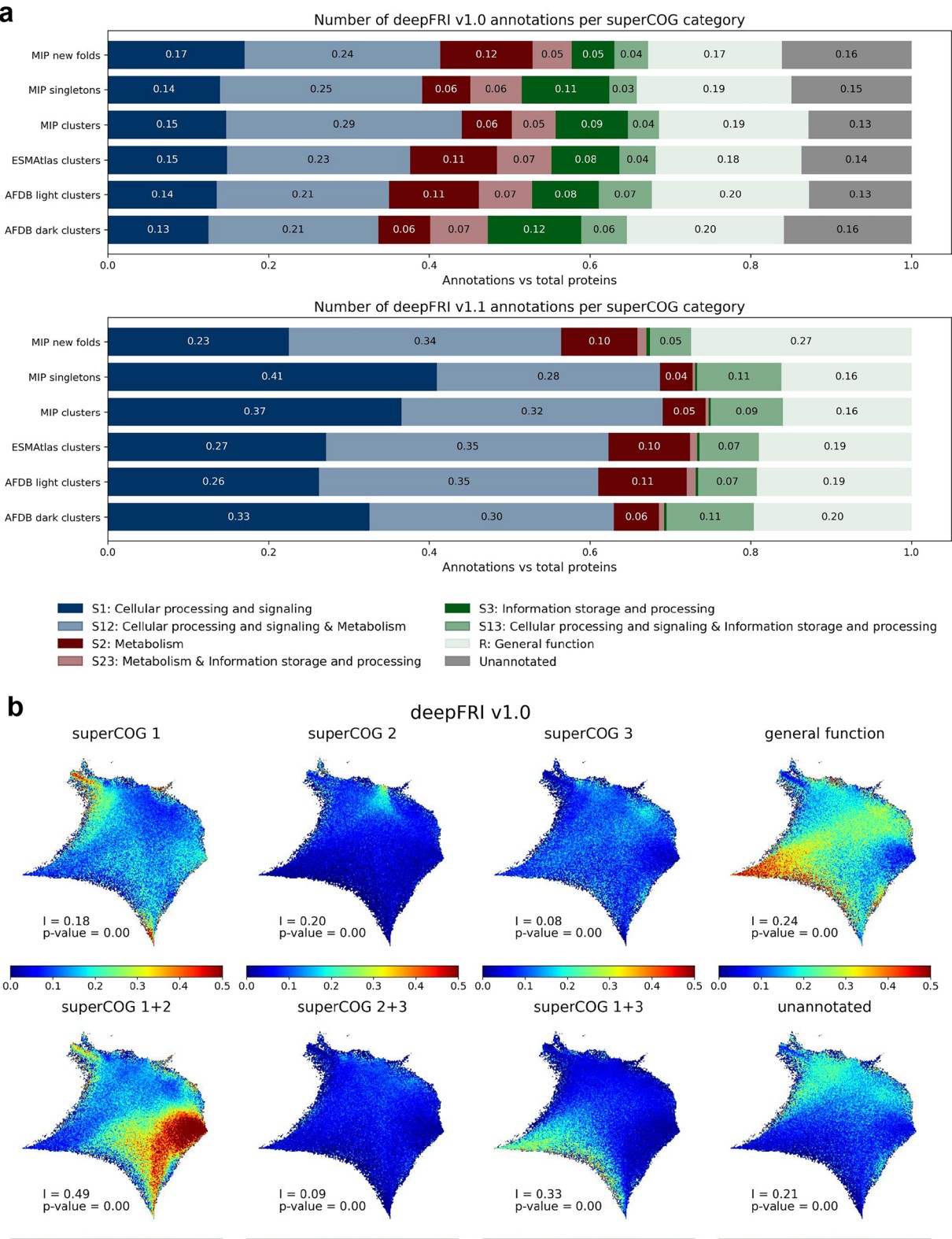

**Fig. 3 | Distribution of functional categories within datasets and the structure space. a** deepFRI annotation of high-quality structural models in each database based on superCOG aggregation. For deepFRI v1.0 and v1.1, annotations with score ≥0.3 and ≥0.5 were considered, respectively. **b** Normalized contribution of each superCOG category for deepFRI v1.0 (see Supplementary Fig. 41 for deepFRI V1.1 predictions). All panels have been generated using normalized Geometricus vectors for all ~1.5 million cluster representatives + singletons (versions showing absolute values as well as based on unnormalized Gemetricus representations can be found in the Supplementary Information – Supplementary Figs. 42, 43 and Supplementary Figs. 44–47, respectively). Moran's I coefficient and its corresponding *p*-value for two-sided statistical test are displayed in each panel. No adjustments were made for multiple comparisons (see Spatial autocorrelation testing paragraph in Methods section for further details).

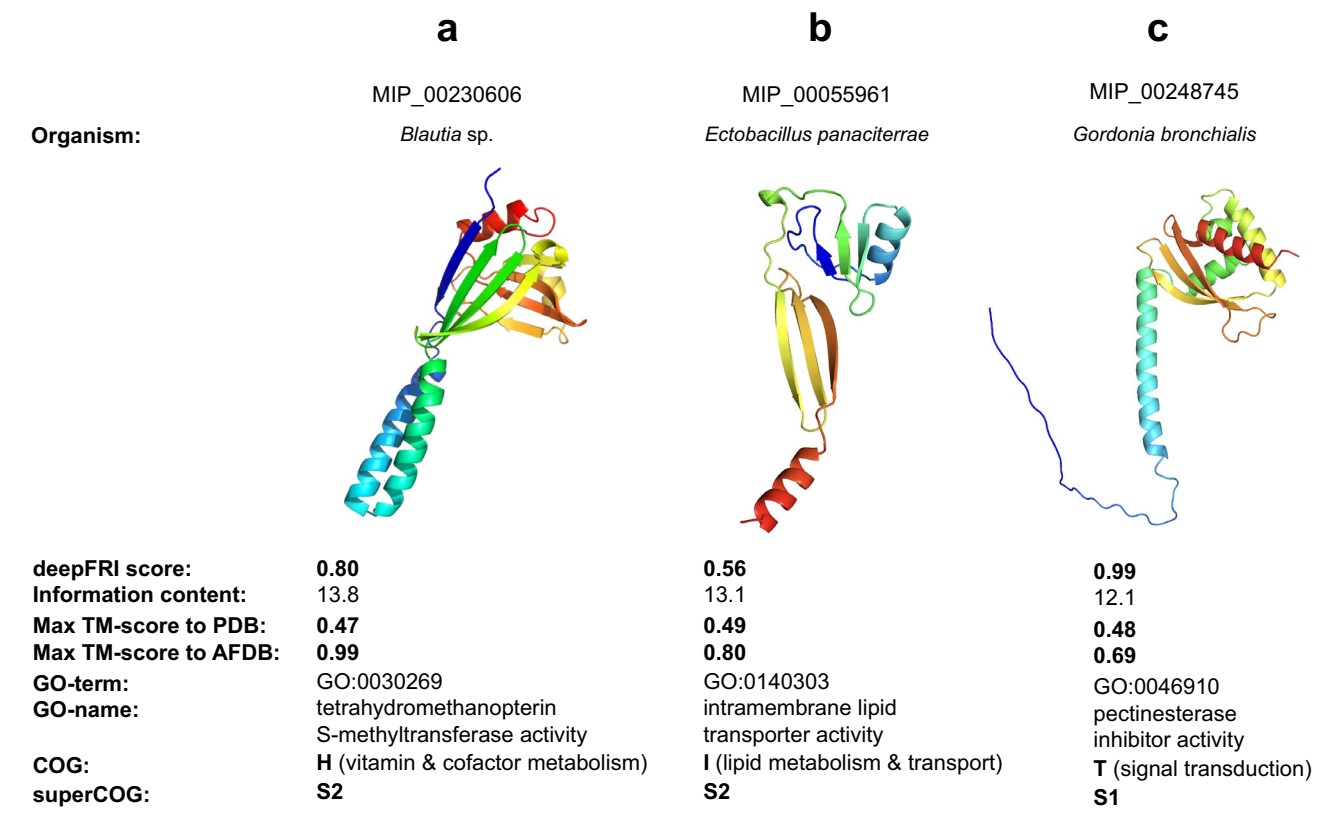

| | **a** | **b** | **c** |
|---|---|---|---|
| | MIP_00230606 | MIP_00055961 | MIP_00248745 |
| **Organism:** | *Blautia* sp. | *Ectobacillus panaciterrae* | *Gordonia bronchialis* |
| **deepFRI score:** | **0.80** | **0.56** | **0.99** |
| **Information content:** | 13.8 | 13.1 | 12.1 |
| **Max TM-score to PDB:** | **0.47** | **0.49** | **0.48** |
| **Max TM-score to AFDB:** | **0.99** | **0.80** | **0.69** |
| **GO-term:** | GO:0030269 | GO:0140303 | GO:0046910 |
| **GO-name:** | tetrahydromethanopterin S-methyltransferase activity | intramembrane lipid transporter activity | pectinesterase inhibitor activity |
| **COG:** | **H** (vitamin & cofactor metabolism) | **I** (lipid metabolism & transport) | **T** (signal transduction) |
| **superCOG:** | **S2** | **S2** | **S1** |

**Fig. 4 | MIP novel fold case study.** Functional overview of MIP novel folds. deepFRI v1.1 has been used for function annotations. **a** AlphaFold2 prediction of MIP_00230606 protein. **b** AlphaFold2 prediction of MIP_00055961 protein. **c** AlphaFold2 prediction of MIP_00248745 protein.

database (TM-score = 0.99, Fig. 4a), while alignments with other known tetrahydromethanopterin S-methyltransferases yielded poor matches (TM-score = 0.47).

Another example (Fig. 4b) involves intramembrane lipid transporters, a diverse group of proteins that move lipids within the membrane. These proteins exhibit structural diversity, with transmembrane domains ranging from a few to over a dozen, and they may function as flippases, floppases, or scramblases[52,53]. This diversity is further influenced by the unique lipid profiles of different cell membranes. The MIP_00055961 sequence was derived from the genome of *Ectobacillus panaciterrae* DSM 19096, a recently classified species with a distinct cellular fatty acid profile compared to other *Bacillales*. Its closest structural match (TM-score > 0.8, despite only 20% sequence similarity) is a RRXRR domain-containing protein from the cyanobacterial organism *Symploca* sp.

The third example (Fig. 4c) concerns pectinesterase inhibitors (PMEIs), extracellular enzymes that inhibit pectin methylesterases (PMEs) and regulate cell wall modification. PMEIs can act against endogenous PMEs or target enzymes from other species in symbiotic or host-pathogen interactions, leading to significant structural variability[54–56]. MIP_00248745, derived from *Gordonia bronchialis* DSM 43247, a known opportunistic human pathogen, might function as a PMEI, potentially influencing glycoprotein formation in human lung mucus. This protein shows high similarity to a protein from *Gordonia insulae* (TM-score = 0.69, with 48% sequence similarity), as well as proteins from *Nocardia brasiliensis* and *Mycobacterium tuberculosis*, to which *G. bronchialis* is related.

## Protein space is dominated by heterogeneous structural clusters

Figure 5 provides an overview of cluster heterogeneity across different databases. Panel a shows the total number of models in each database,

categorized into singletons, homogeneous clusters, and heterogeneous clusters. A significant portion of AFDB clusters is of low or moderate quality (pLDDT ≤70), with 38% for AFDB light and 53% for AFDB dark (red bars). This can be attributed to the structural redundancy within AFDB, which was reduced using Foldseek, leaving behind high-quality protein building blocks and many poorly predicted or intrinsically disordered proteins (see also Supplementary Fig. 18). As a result, these lower-quality models overshadow the high-quality AlphaFold models in the complete AFDB set.

Interestingly, when AFDB models with low pLDDT are discarded, a similar proportion of singletons, homogeneous clusters, and heterogeneous clusters for both AFDB and ESMAtlas is observed. Moreover, most AFDB and ESMAtlas structures combine to form heterogeneous clusters (Fig. 5a). Among all high-quality AFDB structures that cluster with ESMAtlas (or with both ESMAtlas and MIP), only 13% (or 11%) are annotated as dark. In contrast, 31% of all clustered AFDB structures are dark, dropping to 26% when considering only models with a mean pLDDT >70. These observations quantitatively support the conclusions from the structure space analysis in Fig. 1c, indicating that there is a greater overlap between light AFDB and ESMAtlas compared to the overlap between dark AFDB and ESMAtlas. Heterogeneous clusters composed of ESMAtlas and AFDB light proteins tend to be larger than those formed from ESMAtlas and AFDB dark proteins, with median sizes of six and three, respectively. The ten largest heterogeneous ESMAtlas and AFDB light clusters contain between 444 and 714 proteins and are predominantly alpha-helical. Nine out of ten representative proteins are from ESMAtlas and originate primarily from marine metagenomes. Functional predictions from DeepFRI show enrichment in biological process (BP) and cellular component (CC) ontology categories, including functions such as establishment of localization, membrane protein complex assembly, cell periphery organization, and broader

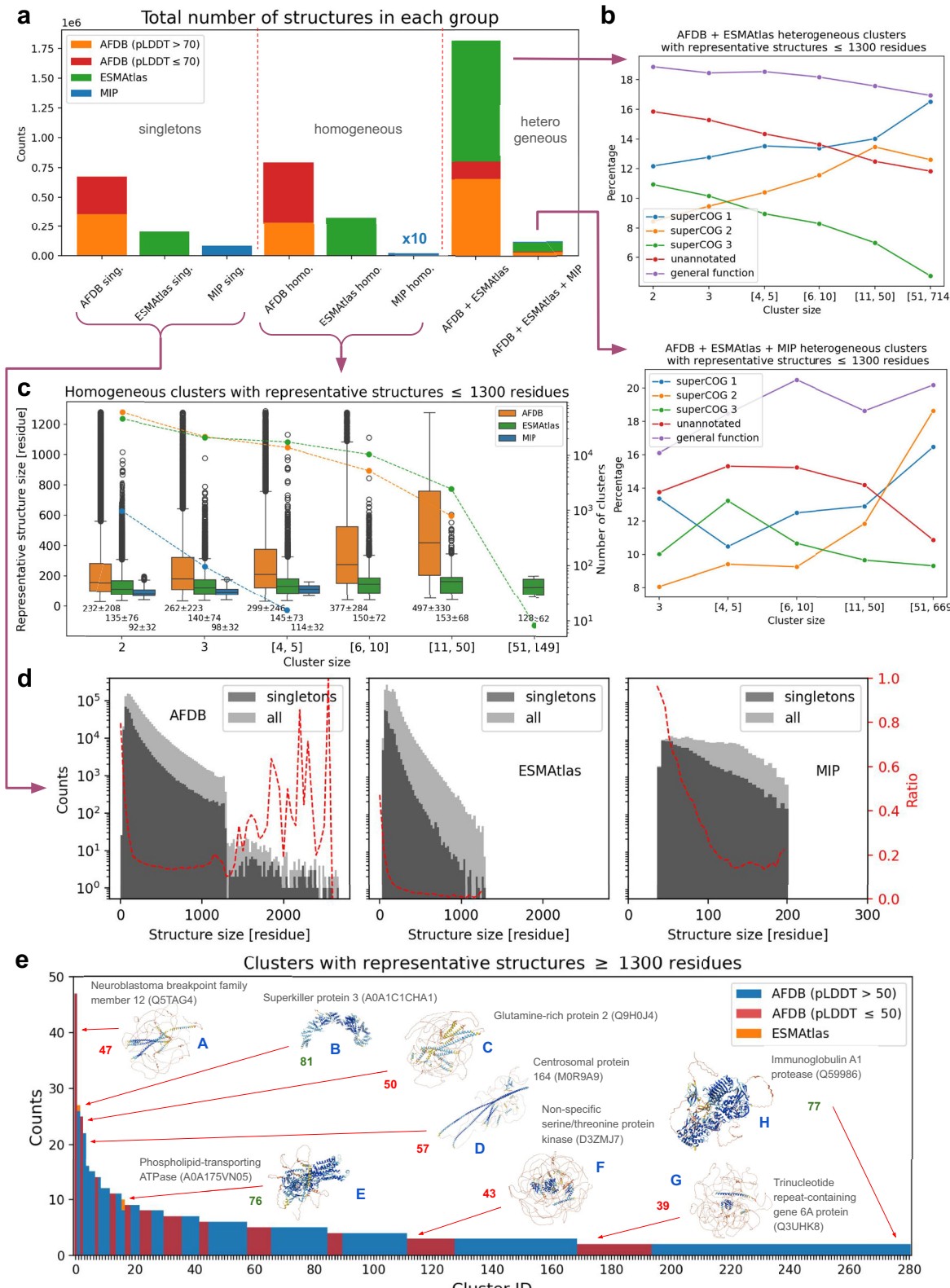

cellular component organization. In contrast, the ten largest heterogeneous ESMAtlas and AFDB dark clusters are smaller – ranging from 37 to 74 proteins – and exhibit far greater structural diversity, encompassing alpha, beta, and alpha-beta classes. All representative proteins in these clusters come from ESMAtlas and originate from a mix of environments – host-associated, engineered, and freshwater metagenomes – and are associated with functions spanning all Gene

Ontology categories, including nucleus, ribosome, cell periphery, plasma membrane, and DNA binding.

Panel b shows that in heterogeneous clusters, the percentage of superCOG 3 proteins decreases with cluster size, while the percentages of superCOG 1 and 2 proteins increase. This diversity can be explained by the fact that superCOG 1 and 2 proteins are subject to fewer constraints than superCOG 3 proteins, whose structures are largely

**Fig. 5 | Cluster heterogeneity. a** Number of structures (clusters + singletons) stratified by database type and cluster group (singletons, homogeneous and heterogeneous clusters respectively). AFDB structures have been split into low- and high-quality groups. **b** SuperCOG category of representative structures vs cluster size for the two largest heterogeneous cluster groups i.e. AFDB + ESMAtlas and AFDB + ESMAtlas + MIP (corresponding plots for homogeneous clusters as well as for all structures building highly functionally homogeneous clusters may be found in Supplementary Figs. 49–51). **c** Dependance between representative structure length and cluster size for homogeneous clusters (corresponding plots for heterogeneous clusters may be found in Supplementary Figs. 52–55). Data are presented as median values with interquartile range (IQR); whiskers indicate variability outside the upper and lower quartiles, and circles represent outliers. Number of clusters with a given cluster size (data points for each bin) is plotted as lines on the right vertical axis. Below each box is the mean ± standard deviation. **d** Number of singletons for each database type versus structure length. Ratio between the number of singletons and the total number of structures is depicted in red (see right vertical axis). **e** Distribution of clusters (mostly coming from the AFDB with two exceptions i.e. ESMAtlas structures depicted in orange) with long representative structures (≥1300 residues). Examples of AFDB structures are also displayed (see also Fig. 2). The number next to each structure represents the mean pLDDT score: values ≤ 50 are shown in red, and those >50 in green. In (**b**–**d**) only high-quality structures have been considered. In (**b**) and (**c**) only structure representatives shorter than 1300 residues have been included.

determined by nucleic acid and ribosome conformation. Conversely, the increase in superCOG 3-related proteins in homogeneous ESMAtlas data (Supplementary Fig. 51) is likely due to the presence of phage/viral proteins derived from metagenomic samples. Additionally, the percentage of unannotated proteins decreases with cluster size, indicating that larger clusters are better characterized functionally. The percentage of general functions also varies between AFDB + ESMAtlas clusters and AFDB + ESMAtlas + MIP clusters, partially influenced by mixed superCOG categories (not shown in the plot) that may express more general annotations. In this analysis, only cluster centroids were compared, so we also checked if the trends hold for functionally homogeneous clusters – and they do (see Supplementary Fig. 49). Similar behavior is observed in homogeneous clusters (see Supplementary Figs. 50, 51), though changes in superCOG 2 are less pronounced. These observations suggest that while all databases are large enough to display similar functional potential at a general level, there is still considerable diversity within specific clusters.

In Panel c, we see that for AFDB homogeneous clusters, there is a positive correlation between cluster size and representative structure length. This trend is not observed in other databases, although a weaker correlation exists for heterogeneous clusters containing AFDB structures (see Supplementary Figs. 52–55 for details). Panel d shows that the percentage of singletons rapidly decreases with increasing structure size, regardless of the database. In AFDB, a significant portion of long protein tails (>1300 amino acids) fail to cluster with others, likely due to a higher density of low-quality predictions and intrinsically disordered proteins, as well as statistical bias (e.g., the scarcity of long structures in ESMAtlas and AFDB). Finally, Panel e illustrates the number of structures in clusters composed of long proteins, which almost exclusively come from AFDB. Most of these clusters are small, and only 31% have centroids with a mean pLDDT higher than 70. The reason for the larger proteins in AlphaFold clusters could be the variation in organisms across datasets: ESMAtlas (and MIP) consist of prokaryotic proteins, whereas AFDB, based on UniProt, also includes eukaryotic proteins. As prokaryotic proteins are generally shorter than the eukaryotic counterparts across all functions, the higher proportion of eukaryotic structures in the AFDB could result in such bias[57].

3D structure visualizations of the largest representatives, forming both homogeneous and heterogeneous clusters, along with functional annotation coverage of different cluster types, are presented in Supplementary Figs. 58, 59, accompanied by a detailed discussion. An interactive version of Supplementary Fig. 59 is available on the paper's GitHub repository (see Data Availability section).

### Taxonomic imbalance in structure databases may cause a predictive bias

The taxonomic composition across our combined datasets is illustrated in Fig. 6, with panels a and b representing AFDB + MIP and ESMAtlas respectively. Our analysis reveals several key insights regarding taxonomic distribution and prediction quality.

Both AFDB and MIP provide taxonomic assignments, with bacterial structures being the most ubiquitous for AFDB light, AFDB dark

and MIP (Fig. 6a). ESMAtlas is based on metagenomic datasets (Fig. 6b), therefore the precise taxonomic assignments can be challenging, thus biome characteristics offer an alternative, with environmental and aquatic biomes comprising the most structures with ESMAtlas.

An interesting pattern emerges in Fig. 6c – functional annotations of structures having the highest taxonomic diversity (entropy) reveal kinase and phosphate transferase activity in the metagenomic (both environmental and engineered biomes), which is expected, given that both of these functions are ATP-dependent, thus playing a key cellular function. Other functions with the highest taxonomic diversity are related to cytoskeleton (dynein) and protein interactions (ankyrins), which also highlights their universal role across organisms.

The percentage of prokaryotic AlphaFold proteins varies significantly between heterogeneous clusters. Specifically, clusters with minimal ESMAtlas content contain fewer prokaryotic proteins, while those with substantial ESMAtlas representation show higher prokaryotic representation. This pattern suggests a non-random distribution of taxonomic groups across structural clusters (see also Supplementary Figs. 60, 61 and Supplementary Tables 6, 7).

Our examination of the ten largest heterogeneous clusters reveals a strong bacterial dominance, with nine clusters primarily composed of bacterial proteins. Only one of these major clusters contains a substantial representation of plant and fungal proteins, highlighting the taxonomic imbalance in current structural databases.

AlphaFold predictions demonstrate higher accuracy for prokaryotic proteins (bacteria and environmental samples) compared to eukaryotic counterparts (see Supplementary Fig. 62). This quality difference likely stems from the greater prevalence of disordered regions in eukaryotic proteins, which are inherently more challenging to predict accurately. Within dark AFDB clusters – those that are less well-characterized – the prediction reliability for eukaryotic proteins is particularly diminished.

### Web server allows for an interactive exploration of the protein universe

Figure 7 illustrates the core features of the interactive web server. Users can search for structures by ID or click on them directly. Once a structure is selected, detailed information is provided, including SuperCOG predictions from deepFRI v1.0 and v1.1, protein length, and database origin, along with a 3D visualization. The server supports filtering by database origin, SuperCOG category, protein length, and AFDB pLDDT, which may be particularly helpful for identifying disordered proteins. For instance, in Fig. 7, structures from all databases were selected, but only high-quality AFDB models (pLDDT >70) with a length <1000 aa, and SuperCOG 1 annotations from deepFRI v1.0, were shown.

### Discussion

This study attempts to grapple with a reality in which high-quality protein 3D structure models have become a commodity. In 2021–2022 with the releases of AlphaFold Database (AFDB),

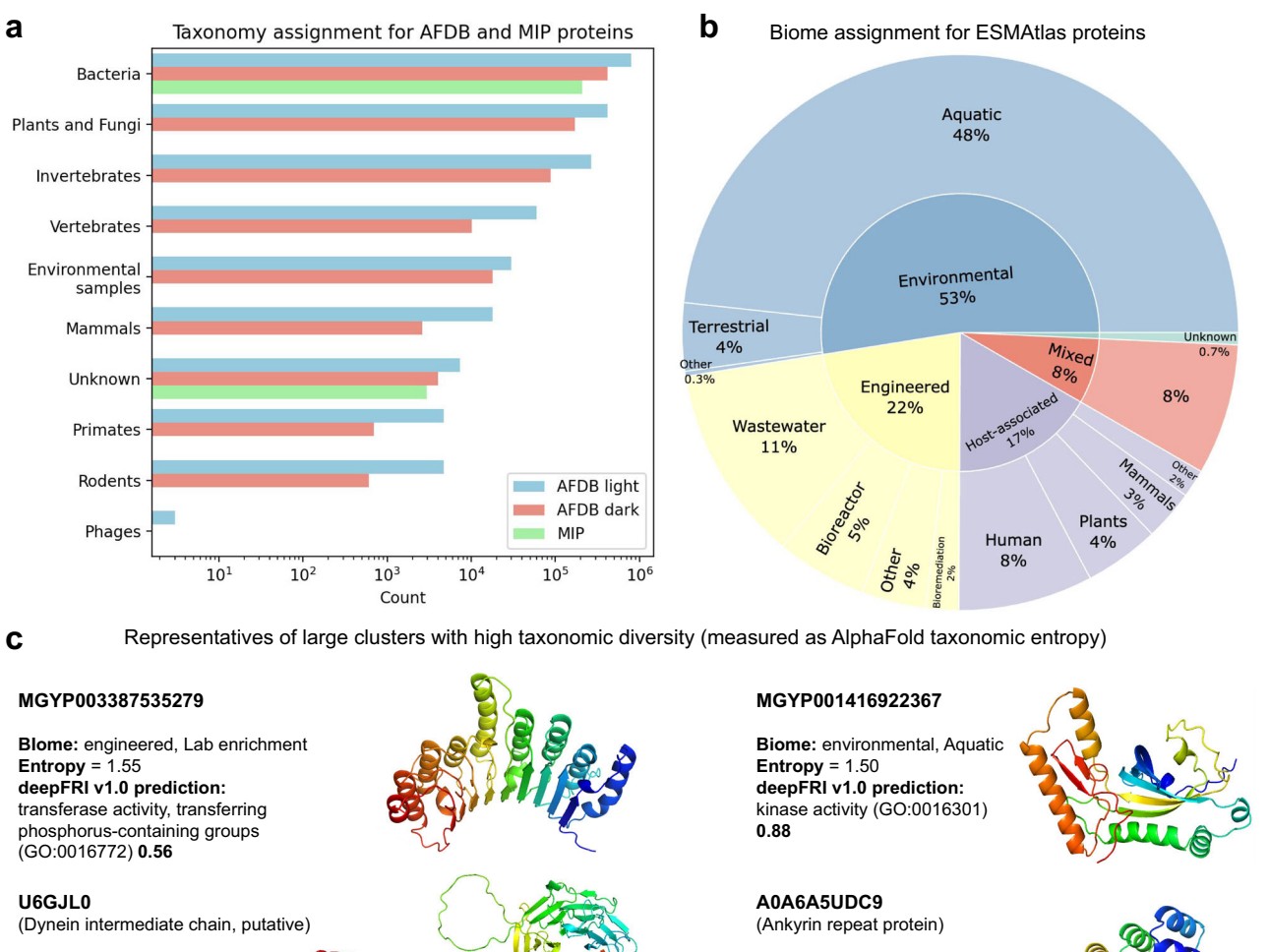

**Fig. 6 | Taxonomic composition. a** Taxonomy assignment for all clustered AFDB (~2,3 million) and all MIP (~200,000) proteins (see Supplementary Table 2). **b** Hierarchical (up to second category) biome assignment for all clustered ESMAtlas (~1,6 million) proteins. **c** Representative structures from four large clusters (size ≥100) exhibit high taxonomic diversity. Entropy has been computed from taxonomic division vectors of AlphaFold-predicted proteins (see Supplementary Tables 6, 7 and "Taxonomy" paragraph in Methods section for details).

ESMAtlas, and other resources, the research community went from a reality in which there were ~200,000 s of high-quality protein structures available, to a reality in which around 1 billion models are available (10,000× change). The field had decades to learn, adapt and develop a deep understanding and best working practices to be dealing with protein sequences at a scale. The rapid technological shift and abundance of 3D protein structure data occured left a conceptual gap in how we deal with, contextualize, or learn from this newly available perspective on biology. Our study offers an outlook on how this can be achieved and proves that in fact it is possible to draw biologically meaningful conclusions from a protein structure landscape perspective.

It has been already demonstrated that the protein structure space is continuous and that the Protein Data Bank contains sufficient structural information to understand the protein structure landscape[17,28–30,58]. This statement is supported by the fact that Alpha-Fold, trained exclusively on PDB structures, succeeded in predicting structures for the vast majority of the UniProt proteins. In this work, we show that when considering clusters of structures derived from genes coding for proteins in the largest available protein structure databases,

the structure space is highly diverse yet protein conformations display gradual shifts across the space, resulting in a continuous structural landscape (Fig. 1). We also observe clear complementarity between these databases: AFDB and ESMAtlas clusters, although greatly overlap, cover different locations of the structure space on a similar level (Fig. 1c). MIP domains (clusters and singletons) complement ESMAtlas cluster representatives but we might speculate if inclusion of ESMAtlas and AFDB singletons would change the picture. At this point, we decide not to include singletons in the analysis due to computational and analytical constraints, though this remains a promising direction for future work. Another potential avenue is to split all input structures into domains, evolutionary conserved structural building blocks, before clustering. This approach would provide a different perspective on the protein structure space, highlighting similarities between AFDB and ESMAtlas at the domain level, which is already covered by the TED resource[18,20]. However, we confirmed that the extension of the protein universe constructed on AFDB, ESMAtlas, and MIP to also cover viral (BFVD database) or artificial (ProtGPT2) structures proved to maintain the overall structure of the space (see Supplementary Figs. 33–37 and Supplementary Tables 3–5).

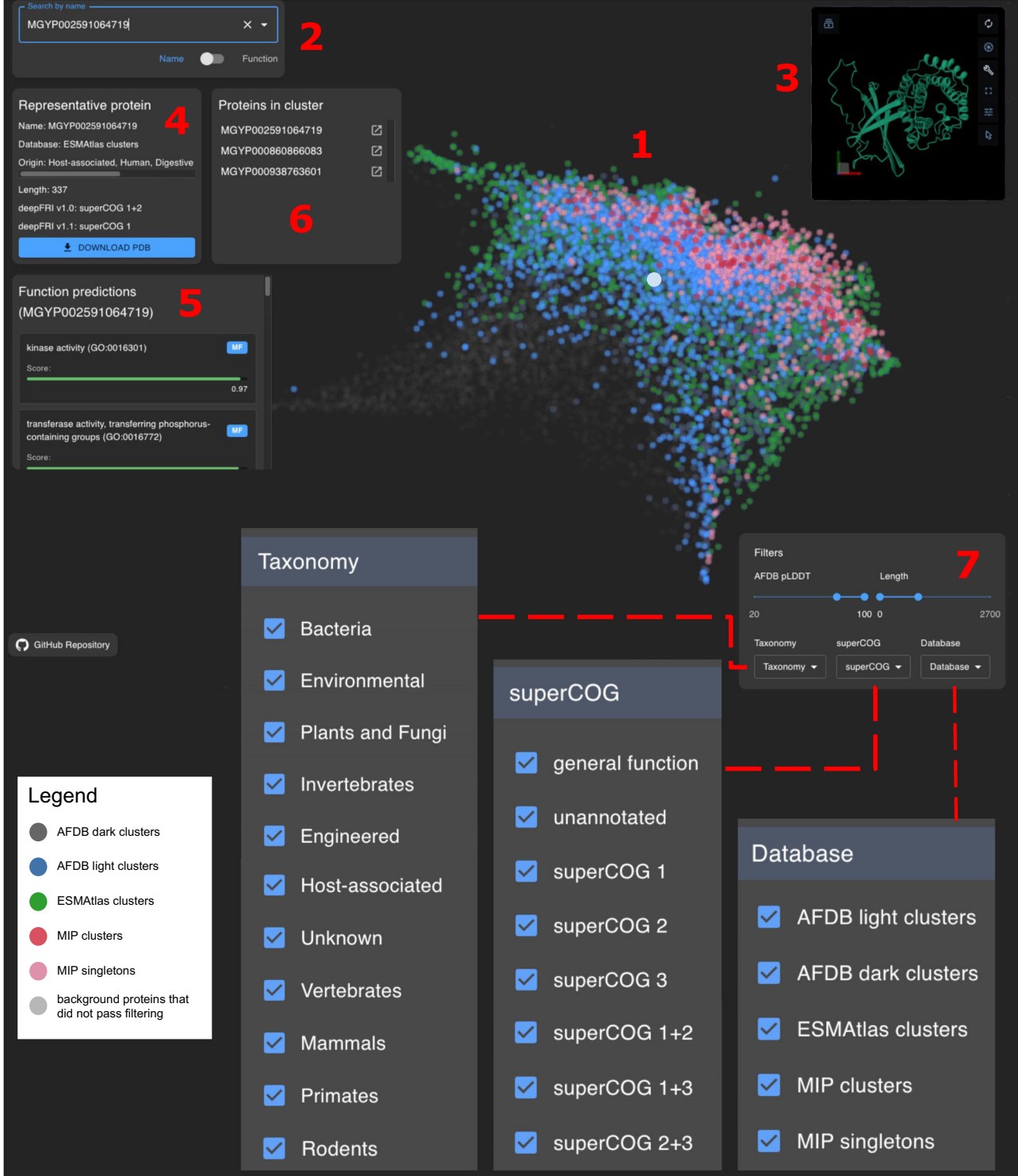

**Fig. 7 | Web server interface.** The central panel (1) displays representative protein structures (~1.5 million non-redundant clusters) as colored points, with each color representing a different dataset (see legend). Users can search for proteins by name (supporting all ~4 million IDs, including novel MIP folds) or by function (2), or select them directly via mouse click. Selecting a protein triggers the display of a 3D structure in PyMOL (3) and an information panel (4) showing details about the representative protein. In panel 5 deepFRI v1.0 function predictions are shown. All proteins within a selected cluster (represented as a single point) are displayed in panel 6, where each protein name can be clicked to view its 3D visualization and function predictions. Users can filter AFDB structures by pLDDT score, and all datasets can be filtered by sequence length, taxonomy, superCOG category, or database type (7).

Functional analysis revealed that all considered datasets possess similar high-level functional profiles (on superCOG level – Fig. 3a). Nevertheless, we can observe certain differences on a cluster level (see Fig. 5b). It appears that novel folds do not necessarily indicate unknown functions (Fig. 4). To accurately determine their functional capabilities, it is essential to utilize advanced annotation tools that do not rely solely on homology. deepFRI offers a great advantage to homology-independent functional annotation, which was demonstrated in a number of papers[21,22,33,34,59,60]. Furthermore, the differences between deepFRI v1.0 and v1.1 create a complementary advantage in

analyzing large databases. While version 1.0 appears more reliable and produces more balanced predictions, version 1.1 proves particularly valuable for studying novel folds (Fig. 4).

Heterogeneity analysis shows that most of the high-quality AFDB and ESMAtlas structures build heterogeneous clusters (with at least one representative from each database), meaning that there is a large structural similarity between them (Fig. 5a). Nonetheless, we can still identify a substantial number of AFDB and ESMAtlas singletons and homogeneous clusters. Due to disparity in how the two main structural databases have been constructed, direct taxonomic comparison within the protein space is difficult. It is apparent, though, that the majority of protein structures currently available represent prokaryotic organisms, predominantly bacteria.

By design, we included all AFDB clusters created in ref. 21. This approach contrasts with the highquality_clust30 definition but enables us to estimate where low-pLDDT AlphaFold predictions – either low-quality models or intrinsically disordered proteins (IDPs) – may reside in the structure space. As expected, deepFRI primarily annotates these as having a general function (Fig. 3b). However, considering that many of them are likely disordered (see refs. 61–64 for correlation between AlphaFold pLDDT and disorder), this highlights the need for improved functional annotation of intrinsically disordered proteins – IDPs usually fall into the superCOG 1 group, given their function as molecular chaperones[65,66]. Given that deepFRI is a structure-based tool, the annotation of IDPs would require a development of novel approach, e.g. based on dynamic modeling of IDPs under specific physiological conditions.

There are some limitations to the study and approach taken by us. Due to the volume of data, we were unable to comprehensively address intrinsic disorder in proteins (except approximating it using pLDDT), and phenomena which are associated with disordered proteins, such as fold-switching, moonlighting or preferred conformations[67–70]. Accurate characterization of disordered regions typically relies on MSA-based methods that are computationally intensive and beyond the scope of our current framework. Additionally, since we used non-redundant sequences as inputs, we are unable to directly examine the sequence-structure relationship, which could, for instance, offer valuable insights when exploring taxonomic diversity. Moreover, the tools used for filtering and clustering (such as Foldseek and Geometricus) introduce limitations in sensitivity, potentially missing finer-grained structural relationships. The use of 2D embeddings, while valuable for visualization, also constrains the complexity of structural patterns that may be captured. Nevertheless, within these methodological bounds, our analysis offers a meaningful and informative view of the protein structure space.

In conclusion, this study provides an analysis of the protein structure landscape by identifying representative structures from three large resources: the AlphaFold database (AFDB), a high-quality subset of ESMFold (called highquality_clust30) and the Microbiome Immunity Project (MIP) database. Our findings indicate that those databases are complementary in their coverage of protein folds (structural complementarity), although their high-level functional annotations (split into three large functional families called Super-COGs) display similar profiles. Constructing a low dimensional representation of cluster representatives from all databases, we offer a detailed look into how proteins are distributed across the structure space. We show that proteins representing all three high-level functional groups, SuperCOGs, are localized in specific regions of the structure space (functional locality). We also provide a detailed analysis of cluster heterogeneity and taxonomic assignments, showing quantitatively to what extent the databases are similar. Additionally, we have developed a web server for personalized exploration of the clustered dataset, facilitating more tailored analyses (Fig. 7).

Ultimately, this work enhances our understanding of the current state of protein structures and their evolutionary and functional implications, offering valuable insights for future research.

## Methods

### Datasets

We considered three large protein structure datasets: AFDB50 i.e. AlphaFold-database clustered at 50% sequence identity level[21], high-quality_clust30 i.e. high quality subset of ESMAtlas (see details here: https://github.com/mosamdabhi/esm_meta_protein/blob/main/scripts/atlas/README.md) with pTM and pLDDT > 0.7 clustered at 30% sequence similarity level, and Microbiome Immunity Project (MIP) (which already consists of structures clustered at 30% sequence identity level)[25]. The structural models for each dataset were retrieved on February 23, 2024. They comprise predictions from AlphaFold[7,25], ESMFold[16], and Rosetta[16,71] respectively. The AFDB models used in this study are from version v4 and follow the naming convention: AF-UniProtID-F1-model_v4. By default, we did not filter the AFDB structures by pLDDT as we wanted to observe the distribution of structures with low pLDDT within the structure space (note that some of them may be intrinsically disordered proteins). However, quantitative conclusions are drawn for AFDB structures with mean pLDDT > 0.7 (to be in line with highquality_clust30 construction) i.e. Figs. 1c, 3a, and 5. We also provide additional plots in the Supplementary Information where the aforementioned constraint is imposed (e.g. the whole heterogeneity analysis section). Similarly to ref. 21 we exclude highquality_clust30 singletons. We did include MIP singletons since they may contain important but rare domains from the tree of life coming from a curated set of genomes. See the Supplementary Information for more details. The ProtGPT2 database was obtained from https://huggingface.co/nferruz/ProtGPT2 (last accessed on February 25, 2025), and the Big Fantastic Virus Database (BFVD) was downloaded from https://bfvd.steineggerlab.workers.dev (last accessed on February 25, 2025).

### Structural clustering

Structural clustering has been performed using Foldseek[10] version e99047b2214b20b7c93f36fc0d54394c843c046b. We followed a similar procedure as the one described in ref. 21. Structure comparison that outputs TM-score has been done using US-align[21,72] version 20230609. Parameter tuning and more details may be found in the Structure clustering section in the Supplementary Information.

### Structure space construction

Structure embeddings have been generated using Geometricus[13,35] v0.5.0 with the ShapemerLearn class that outputs 1024-dimensional shape-mer representation (see https://turtletools.github.io/geometricus/getting_started.html for details). For our purposes, we opted for a motif-level shapemer representation, which aligns more directly with the goals of our analysis. Other methods, such as Foldclass[73], are better suited for domain-level encoding, while FoldToken[74] is less interpretable due to its fully deep learning-based design. We computed shape-mers for all representative structures in the final clustered database comprising 1,505,141 proteins (see Fig. 1 and Supplementary Table 2). The Geometricus representation has been reduced into two dimensions using the Pairwise Controlled Manifold Approximation Projection (PaCMAP)[36] being the current state-of-the-art technique for this purpose. We performed an extensive grid search to tune PaCMAP parameters (see the Structure space section in the Supplementary Information for details). By default, in order to reduce the structure length bias we normalize Geometricus representations, but we also provide results for unnormalized vectors (see Supplementary Figs. 29–32, 44–47). 3D protein structure visualizations have been generated with PyMOL 2.5.0 Open-Source, 2022-11-16.

## Function prediction

Function prediction has been done using deepFRI[27] with release v1.0 [https://github.com/flatironinstitute/DeepFRI] and release v1.1 [https://github.com/bioinf-mcb/DeepFRI]. deepFRI v1.1 has been trained using exactly the same architecture and training procedure as v1.0 but with a different training and validation sets (see the "Functional annotation" section in the Supplementary Information for details). For each Gene Ontology (GO), Information Content (IC) values were pre-computed using the same reference, to allow comparability for different databases.

## go2cog mapping

Mapping of Gene Ontology (GO) terms into Clusters of Orthologous Groups of proteins (COG) has been performed manually with the use of 1227 GO-terms, 26 COGs and 3 superCOGs[27,40,41]. Thanks to GO-term propagation with GO graph (go.obo file available at https://geneontology.org/docs/download-ontology/, access 20 January 2024) we were able to map 4504 GO-terms for deepFRI v1.0 (95% coverage of all outputted GO-terms), 8466 GO-terms for deepFRI v1.1 (91% coverage of all outputted GO-terms), and 9427 GO-terms in total. The mapping is freely available at https://github.com/Tomasz-Lab/go2cog. deepFRI predictions have been mapped to superCOG categories using the following scheme: 1) Use only high-quality deepFRI predictions (threshold put on deepFRI score equals 0.3 and 0.5 for v1.0, and v1.1 respectively – see the Supplementary Information for details). 2) Group predictions by protein. 3) If only R (general function) annotations are present, annotate a given protein as R, but If more than one superCOG category appears, assign them all equally, ignoring R (if present). 4) If all 3 superCOG categories are present, assign them to the R category. For COG categories, deepFRI predictions have been mapped using the same scheme but in point 4) we relaxed the number of different categories from three to at least five.

## Spatial autocorrelation testing

Moran's I coefficients and corresponding $p$-values (presented in Fig. 3b and in Supplementary Figs. 41, 44–45) were computed using the moran.Moran class from the esda Python package (version 2.7.0). The nearest neighbor weight matrix was generated with the weights.KNN class from the libpysal package (version 4.13.0), using 10 neighbors by default ($k = 10$). The results are robust to this parameter choice, as varying k from 1 to 20 yielded consistent test statistics and $p$-values.

## Taxonomy analysis

Taxonomy assignment for AFDB and MIP entries was carried out using NCBI taxonomy mappings obtained from NCBI's FTP site https://ftp.ncbi.nlm.nih.gov/pub/taxonomy (version 2025-01-14). Taxonomy identifiers were retrieved from the following sources: AFDB cluster repository https://afdb-cluster.steineggerlab.workers.dev and MIP Zenodo repository https://zenodo.org/records/6611431, respectively. For ESMAtlas, which is based on the MGnify peptide database, no official taxonomy mappings were available. Instead, we used biome assignments from https://ftp.ebi.ac.uk/pub/databases/metagenomics/peptide_database repository. Specifically, we filtered mgy_biomes.tsv.gz files located in the 2018_09, 2018_12, 2019_05, 2022_05, 2023_02, and 2024_04 subdirectories, as all were found to contain relevant data. Shannon entropy in Fig. 6c has been computed based on taxonomic division vectors of AlphaFold-predicted proteins (the most appropriate proxy of taxonomic diversity) using stats.entropy from SciPy Python package (version 1.14.1).

## Web server

The web server enables users to interactively explore the protein structure landscape. Features include search by protein name or function, and filters for database source, superCOG group, protein length, taxonomy, and AFDB pLDDT score. The frontend has been created using React [https://react.dev], while backend is based on Python's FastAPI library [https://fastapi.tiangolo.com]. We have visualized the 2D projections of the embeddings using SciChart [https://www.scichart.com]. The backbone of the visualization consists of 10,000 points randomly sampled from the whole database. Each change in filters or screen location prompts a request to the backend, which responds with 1000 points matching the criteria and displays them on the frontend. The code for both parts is available at https://github.com/Tomasz-Lab/2dPointVis. The web server is accessible at https://protein-structure-landscape.sano.science and will be continuously improved.

## Reporting summary

Further information on research design is available in the Nature Portfolio Reporting Summary linked to this article.

## Data availability

Source data required to reproduce all figures in the article and Supplementary Information, lists of clusters and singletons obtained from Foldseek, relevant metadata (such as structure space coordinates, functional annotations, and the number of structures from each dataset), as well as Fig. 6b and Supplementary Fig. 59 in HTML format can be found in our Figshare repository [https://doi.org/10.6084/m9.figshare.27203073]. The data mentioned above (except source data) are also available in our GitHub repository [https://github.com/Tomasz-Lab/protein-structure-landscape] (release v1.0). Unless otherwise stated, all data supporting the results of this study can be found in the article, Supplementary Information, and source data files.

## Code availability

The code required to reproduce our analysis is available in our Figshare repository [https://doi.org/10.6084/m9.figshare.27203073] and GitHub repository [https://github.com/Tomasz-Lab/protein-structure-landscape] (release v1.0).

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

## Acknowledgements

The computations reported in this paper were (in part) performed using resources made available by the Flatiron Institute. The Flatiron Institute is a division of the Simons Foundation. We kindly acknowledge the support of World Community Grid team and the community of volunteers who donated their computational resources to the creation of the MIP database. We gratefully acknowledge Polish high-performance computing infrastructure PLGrid (HPC Center: ACK Cyfronet AGH) for providing computer facilities and support within computational grant no. PLG/2022/015689. Special thanks also go to the high-performance cluster at the Małopolska Centre of Biotechnology of Jagiellonian University, and to its dedicated system administrator, Wojciech Pilch, for his invaluable support in our computational endeavors. We kindly acknowledge the support of Rosetta Commons for J.K.L. P.D.R. is supported by the Simons Foundation. This work is supported by the EU's Horizon 2020 programme (grant no. 857533, Sano) and co-financed by the Polish Ministry of Science and Higher Education (contract no. MEiN/2023/DIR/3796). T.K. and P.S. are supported by the National Science Centre, Poland grant 2023/05/Y/NZ2/00080. W.W. is supported by the Ministry of Science grant no. PN/01/0195/2022.

## Author contributions

P.S. T.K. P.D.R. and J.K.L. designed the original MIP project and this study. P.S. L.S. and T.K. performed analysis. L.S. created the go2cog mapping. P.S. and W.W. retrained deepFRI v.1.1. W.W. developed the web server. P.S. T.K. L.S. J.K.L. and P.D.R. wrote the manuscript. P.S. and T.K. revised the manuscript. All authors read and approved the final version of the article.

## Competing interests

The authors declare no competing interests.
