## [Transparent Peer Review file · Nature Communications]

Large protein databases reveal structural complementarity and functional locality

Corresponding Author: Dr Tomasz Kosciolk

Version 0:

Reviewer comments:

Reviewer #1

(Remarks to the Author)

Review NatComms

“Large protein databases reveal structural complementarity and functional locality”

I'd like to thank the Editor for the invitation and the authors for a great manuscript, which expands what we currently know about the protein universe.

Szczerbiak and colleagues created structural clusters from very large protein databases (AFDB, ESMAtlas and MIP), converted them into latent representations in embedding space and used a cartographic approach, mapping known areas and highlighting portions of space with no current knowledge.

The manuscript is well written, the figures comprehensive and the supplementary materials extensive.

Here are some comments.

Main comments:

- 1) How much of the theoretical sequence/structure space as defined by Levitt is covered by current sequence and structure resources? Would randomly generated sequences from ProtGPT2 place somewhere in these representations?
- 2) A big ask, but with the new BFVD resource by Steinegger and colleagues, it would be interesting to see how viral-centric resources fit into these maps.

Minor comments:

Page 2. References 8 and 9 should be swapped to reflect temporality of the resources.

“In this study, we analyze three large protein structure datasets originating from disting sources. The AlphaFold Protein Structure Database (9)”. Is this the correct version? If it includes the whole of UniProt it should be reference 8.

These data found its .. where they're accessible through text-based web-based?

Why was reference 10 used for clustering?

Reference 21 is now published.

The AFDB is based on a portion of UniProt, not its entirety.

Page 3.

As CATH is a proxy of PDB, the lower coverage found is expected. Does this change if you take into account assignments from the TED resource (ted.cathdb.info), which covers AFDB? It would be really interesting to see how the coverage overlaps or if the segregation between classes is still evident.

Page 4.

Were other tokenization or encoding approaches investigated besides Geometricus? I.e. Foldclass or Foldtoken?

Page 5.

Is there any reason for referencing (14-16)?

The landscape can be a bit biased, as AFDB is limited in most instances to 1280 residues. Only entries from Swissprot or proteomes have structures to up to 2700 residues, therefore the analyses thresholds for over-1300 residues proteins are clearly affected by this. This should be more evident in the manuscript, or if possible, addressed to include either entries with multiple fragments or lowering the cutoff to 1280 residues.

Page 6.

Although I do like the representation at a glance of the structure space with some structures highlighted, I'd suggest removing some examples and putting the side-views of the 'greens' inside a box containing the other view as well. The figure is crowded as it is, and that might help with clarity.

Page 7.

Regarding Figure S33, I'd suggest specifying the equation for Precision in the text and use 'Precision' in the axis. As this is not explained in the text, it might be confusing to some readers.

The definition of Moran's I Statistic could use some expansion on the significance.

I would combine figure 3B and S36 and put it in Supplementary.

Page 9.

The IC content is dataset-dependent, while GO depth is not. The two are not interchangeable. For example, do I always get deeper GO annotations when looking at MIP data? Is the annotation levels deeper on average than proteins from other resources? What are MIP proteins compared to in the first place when annotated with GO?

Page 10.

Ideally filtering should be performed always before clustering, as structural variability caused by low pLDDT structures affects Foldseek clustering quite significantly!

These observations... onwards to the end of the paragraph. I would expand on this as it is an interesting finding.

Page 11.

The conclusions around Panel C are affected by the size limitations of AFDB.

Page 14 and 15.

The conclusion section reads well, but there are no caveats with the work performed, or the effect of pLDDT, IDPs or issues with pLM-based structure predictors. Assuming that the snapshot of structure space is correct and not biased by the limitations of current methods (particularly ESMFold) can be misleading.

Supplementary.

The benchmarking of the parameters used for Foldseek clustered might need a section on single-domain chains, as variability in linker regions seems to be a limitation of full-chain clusters as seen in TED and the AFDB clustering paper in Nature. What's the RMSD within clusters? And is the TMscore cutoff used qtm score, tmscore or alntmscore? A normalised version of qtm score/tmscore is often more accurate than alntmscore. Since limiting the TMscore to the aligned region might cause issues if the alignment is poor or one of the two proteins is a fragment. What was the flag used in Foldseek? Coverage of both query and target (bidirectional)?

Thanks for the great work and the pleasant read! I'm sure the community will benefit from this!

(Remarks on code availability)

The code is accessible and well documented. It includes code as python notebooks to recreate figures in the manuscript.

Reviewer #2

(Remarks to the Author)

In this study, the authors performed a large-scale structural clustering using protein structural models derived from three databases: AFDB, ESMAAtlas, and MIP (Microbiome Immunity Project). Using the Foldseek program for clustering, they constructed a structural space that contains a single representative for each cluster derived from the individual databases. The principal structural conclusion of this extensive comparison is that the three databases exhibit structural complementarity, with AFDB and ESMAAtlas displaying significant overlap within the structural space. In contrast, the contribution of MIP proteins to this space is more difficult to interpret, given that MIP predominantly comprises shorter and/or single-domain structures. Additionally, the authors observed that globular or ordered structures or structural motifs change gradually throughout the structural space, and they identified a specific region that likely contains intrinsically disordered proteins (IDPs), primarily represented by lower-quality AFDB models.

Subsequent functional annotation with deepFRI enabled the authors to define groups of structures associated with general functionalities—namely, “Cellular Processing and Signaling,” “Metabolism,” and “Information Storage and Processing.” They noted that proteins assigned to these groups tend to cluster in specific regions, a phenomenon the authors describe as “functional locality.” Additionally, an entire section is dedicated to the putative functional annotation of novel folds, most of which are derived from MIP examples.

Finally, with the obtained data, authors designed a web server to display the obtained information. The web server is both visually appealing and user-friendly.

While I acknowledge the considerable effort undertaken by the authors to execute such a massive analysis of structural similarities across proteins from three distinct databases, I remain concerned about the biological relevance of the conclusions drawn from this work. Specifically, the actual contribution of MIP proteins to the overall structural space delineated by AFDB and ESMAtlas is challenging to assess due to the size and organization of the proteins in MIP. Furthermore, considering that AFDB likely contains millions of homologous proteins that also appear in ESMAtlas, the conclusions regarding overlapping databases and their structural similarities might be inherently biased.

From a biological perspective, it may have been more informative to express the overlapping and differences between AFDB and ESMAtlas in terms of protein classes/taxonomies and also that these differences be available as tables in the web site/supplementary files. Furthermore, same information as taxonomic groups (e.g., eukaryotic, prokaryotic, archaeal, viral) and/or functional categories, could be used to characterize the heterogeneity within a given cluster or structural region more effectively than merely considering the origin of each structural model.

The exploration of the structural space of globular domains associated with intrinsically disordered regions (IDRs) is an intriguing aspect of the manuscript; however, relying solely on average pLLDT values to identify IDRs or IDPs could be problematic and should ideally be complemented with additional disorder estimation methods.

Regarding the functional analysis and the notion of “functional locality” described by the authors, one must consider whether this observation is simply a consequence of homologous proteins naturally occupying similar regions within the structural space. In this context, the pertinent null hypothesis is: What additional information does structural data provide in revealing similarities between proteins? In other words, would clustering all these proteins based solely on sequence data yield comparable results? Although it is well understood that protein structures are generally more conserved than their sequences—thereby potentially improving the detection of remote homologues or similarities—it should be noted that, while Foldseek offers enhanced speed, it is approximately 15% less sensitive than methods such as Dali (Kempen et al., 2023). Could the authors please clarify how they believe that the reduced sensitivity of their method affects the overall findings, particularly in the section devoted to new folds? Additionally, is there any supplementary functional or biological information available that could further support the predicted functions of the examples presented in Figure 4?

Regarding the web server, while it is well designed and visually appealing, I encountered difficulties when attempting to locate the examples mentioned in Figure 4, which would help in understanding the uniqueness of those folds. Moreover, I attempted to download the data from the MIP database; however, it appears that the entire dataset must be downloaded in order to access just those specific examples. Related to the search capacity, Would it be possible to search for any protein contained in a given cluster besides the one being the representative one?

Overall, I consider the manuscript to be a noteworthy piece of work, but I believe it would benefit from addressing the concerns raised above and further elaborating on the biological implications of their findings.

(Remarks on code availability)

Reviewer #3

(Remarks to the Author)

In this study, the authors provide a study of the structural and functional landscape of the protein universe. They collate a large number of predicted structures from AFDB, ESM Atlas, and MIP. They also functionally annotate the proteins using Deepfri, a computational function prediction method. The authors provide results on a web server, which is searchable by UniprotID or function. Overall, there was a large investment in this body of work, but it left me wondering about the utility and novelty of this study. Furthermore, I have identified certain methodological issues.

This work relies primarily on predicted structures and functions. It is not clear if there are any experimentally produced structures (e.g. via X-ray crystallography or NMR) nor experimentally annotated functions. Those can help provide verified structure or function data to what is otherwise a compilation of structural and functional predictions.

Webserver: since the proteins used have been clustered, it means that only a representative of a protein group will be represented in the server. While that is a sound strategy for creating a structural landscape, it makes it impossible to search for a protein using any uniprot ID, since the user needs to know, for example, what is the representative uniprotID chosen for TP53? (I could not find it).

While the authors provide a measure of confidence in structure prediction using PLDDT they do not provide the equivalent function prediction for the DeepFri model, which is a confidence level of 0-1. Furthermore, PLDDT is a per-residue metric, not a per-protein metric. It is not clear that a protein with an overall low PLDDT does not have a reliable structure, it may mean that it just contains many disordered regions, but the non-disordered regions are predicted correctly. Therefore, it is not clear what purpose the PLDDT filter serves.

The webserver provides a static map of the dimensionality reduction of this study. It is zoomable, but not translatable.

Furthermore, I would expect that clicking on any of the dots (the clustering representative) would provide me with information on this particular protein, but that does not happen. Therefore, it is unclear why this mapping is provided in a zoomable/clickable format. (Unless there is a bug, and this function is indeed there, as it seems to be intuitive that it should be).

The dimensionality reduction map purports to provide a variety of protein structure and function space. But without the ability to rotate, translate, and color-code the map by functions or structure.

Finally, I do not understand the novelty of this study, and the value it provides to advancing our understanding of protein structure and function landscape. The authors do not articulate that anywhere, and the only justification I can find was in the abstract and the discussion that this study “lays the groundwork” for future studies. However, it is not said how this study lays the groundwork for future studies.

Minor: The data could use better documentation, e.g. how to open the parquet files provided and a readme file detailing their format

P. 14 "...gene-coding proteins...". genes code for proteins, not vice-versa.

(Remarks on code availability)

The code is not intended to be replicable on another machine, as far as I can tell.

Version 1:

Reviewer comments:

Reviewer #1

(Remarks to the Author)

I thank the authors for addressing in detail all of the points I've raised.

Particularly, for testing the hypothesis on BFVD and ProtGPT2. The figures and text are also more legible.

I read Reviewer 3's comments and in my opinion the authors addressed them satisfactorily.

I have no further comments. Well done!

(Remarks on code availability)

I am content with the status of the code repository.

Reviewer #2

(Remarks to the Author)

I'd like to thank the authors for kindly addressing all my concerns. I believe the manuscript has improved significantly thanks to the modifications made to both the text and the web server.

I have evaluated the authors' responses to Reviewer 3. I believe their answers adequately address the reviewer's concerns.

In the first question, the reviewer asked whether the authors used experimentally based structure–function relationships. The authors responded that they used CATH, which I consider acceptable. The second concern was related to the fact that the web server initially retrieved only cluster representatives, which significantly limited searches for specific proteins (this was also one of my concerns). This issue has now been fixed. The next three points raised by the reviewer referred to missing information and/or functionality on the website — such as DeepFRI and pLDDT filtering scores, and zoom/rotation capabilities — all of which have been addressed. All of the concerns raised by Reviewer 3 were accompanied by appropriate modifications to the manuscript. Finally, the last question was about the biological relevance of their findings (also one of my own concerns), which the authors improved by adding new paragraphs in the introduction and the general conclusions.

Overall, I believe the authors made a strong effort to integrate these databases and to provide both structural and biological insights into this interface. If I had to point out a weakness, it would be the limited biological interpretation of their findings. However, I also believe it is up to the readers to explore potential applications, and it may be beyond the authors' scope to provide a full biological context at this stage — I think their work represents an important first step.

(Remarks on code availability)

Reviewer #1 (Remarks to the Author)

I'd like to thank the Editor for the invitation and the authors for a great manuscript, which expands what we currently know about the protein universe.

Szczerbiak and colleagues created structural clusters from very large protein databases (AFDB, ESMAAtlas and MIP), converted them into latent representations in embedding space and used a cartographic approach, mapping known areas and highlighting portions of space with no current knowledge.

The manuscript is well written, the figures comprehensive and the supplementary materials extensive.

Here are some comments.

Main comments:

1) How much of the theoretical sequence/structure space as defined by Levitt is covered by current sequence and structure resources? Would randomly generated sequences from ProtGPT2 place somewhere in these representations?

Over the years, Michael Levitt published many works addressing the protein universe concept. We are aware of 6 papers by Levitt and colleagues which directly address some facet of the protein sequence and/or structure universe or the fold space. In his 2009 PNAS paper Levitt assesses the coverage of the existing sequence space with structures and the saturation of the 3D structure space with different protein folds. Levitt notes that single domain families were already growing slowly by 2009, while multi-domain proteins contributed to the vast majority of observed increase in diversity. His argument was that almost all novelty in the protein universe comes from the arrangement and combination of known single domains along a multi-domain sequence, rather than the discovery of fundamentally new single domains or folds. Our work focuses on full-chain proteins, hence we are not able to fully evaluate this claim using our own results. But with conjunction with, e.g. TED, we may arrive at a conjecture that the claim does not hold anymore. TED was able to discover close to 7,500 putative new folds. Notably, Levitt's work does not include metagenomics sequences which, as we show in our paper, significantly impacts observations we are able to make. We have added additional information in the Introduction (page 3):

As such, our study goes well beyond prior attempts to characterize the protein universe [28–32]. Unlike many previous studies, here we focus on full-chain structures, showing significant novelty coming from multi-domain proteins, and, previously discarded or unavailable, metagenomics sequences primarily represented in ESMAAtlas.

Regarding ProtGPT2 - this is a great suggestion! We elaborate on this below, addressing the second comment about BFVD, since we treated them in a similar manner.

2) A big ask, but with the new BFVD resource by Steinegger and colleagues, it would be interesting to see how viral-centric resources fit into these maps.

We appreciate the reviewer's suggestion. Both the ProtGPT2 and BFVD datasets are substantially different from those analyzed elsewhere in the paper – ProtGPT2 consists of artificial sequences and BFVD is highly domain-specific. Hence, we initially excluded them from our primary analysis, which focused on biologically grounded and general-purpose structure repositories. However, following the reviewer's input, we conducted a comparative analysis with the datasets used in our study. We have added a summary of this comparison at the end of the section "*Protein conformations display gradual shifts across the structural landscape*", along with a detailed evaluation in the Supplement (see subsection "*Other databases*"; Figs. S33-S37 and Tables S3-S5). Overall, both ProtGPT2 and BFVD structures span the structural landscape relatively evenly. We also observed a high number of singletons, largely correlated with low pLDDT scores. While these datasets highlight exciting potential for expanding the known protein structure space, a full exploration of their biological relevance remains outside the scope of this work.

In Results we have added:

To evaluate the generalizability of our dimensionality reduction approach, we applied it to two additional datasets: a smaller set of 10,000 AlphaFold2 models of artificial sequences generated with ProtGPT2 [38], and a larger set of 351,242 structural models from the Big Fantastic Viral Database (BFVD) [39]. Results are detailed in the Supplement, subsection "Other databases". As shown in Figs. S33-S34, ProtGPT2 structures are distributed fairly evenly across the structural landscape, while viral structures – though spanning the entire space – tend to exhibit a higher alpha-helical content. Notably, our method successfully identified structural outliers, such as short unfolded motifs (right panel, Fig. S34), which are absent in the curated dataset that includes AFDB, ESMAAtlas, and MIP cluster representatives. Interestingly, a substantial portion of both ProtGPT2 and BFVD structures appear as singletons relative to these reference databases (Table S4), but due to their low pLDDT scores (Figs. S35-S36). These findings indicate that our approach generalizes well and is applicable to other protein structure datasets.

Added to Discussion:

However, we confirmed that the extension of the protein universe constructed on AFDB, ESMAAtlas, and MIP to also cover viral (BFVD database) or artificial (ProtGPT2) structures proved to maintain the overall structure of the space (see Figs. S33-S37 and Tables S3-S5).

Minor comments:

Page 2. References 8 and 9 should be swapped to reflect temporality of the resources.

“In this study, we analyze three large protein structure datasets originating from distinct sources. The AlphaFold Protein Structure Database (9)”. Is this the correct version? If it includes the whole of UniProt it should be reference 8.

Yes, absolutely. We corrected this.

These data found its .. where they're accessible through text-based web-based?

Thank you, corrected.

Why was reference 10 used for clustering?

This was a typo - we corrected this.

Reference 21 is now published.

Thank you - we updated the reference.

The AFDB is based on a portion of UniProt, not its entirety.

Thank you for highlighting this. We replaced “based on UniProt” with “based on a sizable part of the UniProt”.

Page 3.

As CATH is a proxy of PDB, the lower coverage found is expected. Does this change if you take into account assignments from the TED resource (ted.cathdb.info), which covers AFDB? It would be really interesting to see how the coverage overlaps or if the segregation between classes is still evident.

We chose to use CATH because it represents an annotated subset of the PDB. Broadly speaking, it captures the structural building blocks of experimentally determined proteins. Our aim was to explore how these domains are positioned within the structural landscape, offering an indirect glimpse into regions populated by multi-domain proteins. This analysis was intended to be illustrative rather than quantitative. Similarly, we opted not to include TED, despite recognizing its value as a resource. For a fair and meaningful comparison, it would be more appropriate to first extract single-domain representations not only from AlphaFold and MIP, but also from ESMAAtlas, before applying TED mappings. To address this point, we added a note to the Discussion section and tied it together with our comments on BFVD and ProtGPT2.

Page 4.

Were other tokenization or encoding approaches investigated besides Geometricus? I.e. Foldclass or Foldtoken?

We selected Geometricus because it is a well-established method, published in *Bioinformatics*, that enables protein tokenization at the motif level. While we are aware of other approaches, including the one highlighted by the reviewer, at the time we were developing our methodology, both were still in preprint form

(<https://www.biorxiv.org/content/10.1101/2024.03.25.586696v1>,
<https://www.biorxiv.org/content/10.1101/2024.07.08.602548v1>). Additionally, FoldClass is more suited for domain-level encoding, and FoldToken is less interpretable, as it is a fully deep-learning based approach. For our purposes, we opted for a motif-level shape-mer representation, which aligns more directly with the goals of our analysis.

In Methods section we have added:

Other methods, such as Foldclass [73], are better suited for domain-level encoding, while FoldToken [74] is less interpretable due to its fully deep learning-based design.

Page 5.

Is there are reason for referencing (14-16)?

We removed these citations in the first paragraph of section "*Protein conformations display gradual shifts across the structural landscape*".

The landscape can be a bit biased, as AFDB is limited in most instances to 1280 residues. Only entries from Swissprot or proteomes have structures to up to 2700 residues, therefore the analyses thresholds for over-1300 residues proteins are clearly affected by this. This should be more evident in the manuscript, or if possible, addressed to include either entries with multiple fragments or lowering the cutoff to 1280 residues.

This bias is inherent to all databases, as clearly shown in Fig. 5D. Our goal was to analyze the full dataset regardless of protein length. However, to mitigate this bias, we stratified Fig. 5 into several panels based on length cutoffs. Moreover, there are only 542 representative proteins longer than 1,300 residues, so their inclusion or removal has a negligible impact on both the clustering and embedding generation steps.

Page 6.

Although I do like the representation at a glance of the structure space with some structures highlighted, I'd suggest removing some examples and putting the side-views of the 'greens' inside a box containing the other view as well. The figure is crowded as it is, and that might help with clarity.

Thank you for this suggestion. We improved visibility of the "greens" and updated Figure 2. We didn't remove any examples since all of them are important for our narrative.

Page 7.

Regarding Figure S33, I'd suggest specifying the equation for Precision in the text and use 'Precision' in the axis. As this is not explained in the text, it might be confusing to some readers.

The definition of Moran's I Statistic could use some expansion on the significance. I would combine figure 3B and S36 and put it in Supplementary.

We thank the reviewer for these valuable suggestions. In response, we have replaced the expression $TP/(TP + FP)$ with the more standard term *precision* in Fig. S38 (note that the figure numbering has changed due to the addition of five new figures in the "Other

databases” subsection). We have expanded the explanation of Moran’s I coefficient in the main text:

(...) Moran’s I statistic – a measure of spatial autocorrelation ranging from 0 (no autocorrelation) to 1 (strong autocorrelation). Our analysis yielded a Moran’s I value indicative of significant spatial clustering, with a p-value near zero, confirming that functionally related categories are not randomly distributed but instead exhibit pronounced spatial autocorrelation (see Fig. 3B and Methods for details).

We also provided additional methodological details in the “Spatial autocorrelation testing” subsection of the Methods. Due to updates in Python dependencies used to generate Fig. 3 and Figs. S41, S44–S45, the Moran’s I values for some SuperCOGs have changed slightly (by 0.01 for SuperCOG 1 and 2+3 in Fig. 3). However, these minor changes do not affect the qualitative or quantitative conclusions of the study. We have also chosen to retain Fig. 3B in the main text, as it directly supports the functional locality argument – one of the central conclusions of our work.

Page 9.

The IC content is dataset-dependent, while GO depth is not. The two are not interchangeable. For example, do I always get deeper GO annotations when looking at MIP data? Is the annotation levels deeper on average than proteins from other resources? What are MIP proteins compared to in the first place when annotated with GO?

While GO depth only considers the position of a term in the gene ontology directed acyclic graph (DAG), IC incorporates the actual usage and distribution of terms in annotation databases. This is particularly valuable when analyzing large datasets (AFDB, ESMAtlas) where the biological relevance of annotations varies significantly. Large datasets often suffer from annotation bias (e.g. approximately 58% of GO annotations relate to only 16% of human genes). IC-based approaches can help mitigate this "rich-getting-richer" phenomenon by quantifying the specificity of terms based on their frequency in the corpus, providing a statistical framework to identify truly informative annotations. Moreover, many annotations in large datasets are shallow in the DAG, representing generic terms without describing particular molecular functions, biological processes, or cellular components. Simple depth metrics would fail to identify this issue, while IC-based approaches can highlight the lack of specific annotations. GO depth fails to capture the semantic context of terms. Two terms at the same depth may have vastly different specificities depending on their position in different branches of the ontology. This is particularly problematic in large datasets where comprehensive functional analysis is required.

As a clarification, for all databases, the IC values were pre-computed using the same reference, hence IC values for different databases are directly comparable. We’ve added the following sentence to the Methods (“Function prediction”):

For each Gene Ontology (GO), Information Content (IC) values were pre-computed using the same reference, to allow comparability for different databases.

with appropriate reference in the first paragraph of section “Exploring functional diversity in novel protein folds”:

(...) (see “Function prediction” in Methods section for more details)

Page 10.

Ideally filtering should be performed always before clustering, as structural variability caused by low pLDDT structures affects Foldseek clustering quite significantly!

Yes, we agree. However, in our approach, we intentionally chose not to exclude low-pLDDT AlphaFold predictions, as they represent a particularly interesting subset. Rather than treating them as outliers to be filtered out, we considered them a meaningful part of the dataset. Our aim was to observe how these structures cluster alongside the rest, and we analyzed them separately – primarily in Fig. 5.

The phrases “filtering after clustering” and “clustering after filtering” were potentially misleading, so we revised the sentence to more accurately reflect the actual procedure:

Interestingly, when AFDB models with low pLDDT are discarded, a similar proportion of singletons, homogeneous clusters, and heterogeneous clusters for both AFDB and ESMAAtlas is observed.

These observations... onwards to the end of the paragraph. I would expand on this as it is an interesting finding.

We thank the reviewer for this suggestion. We have added the following at the end of the paragraph:

Heterogeneous clusters composed of ESMAAtlas and AFDB light proteins tend to be larger than those formed from ESMAAtlas and AFDB dark proteins, with median sizes of six and three, respectively. The ten largest heterogeneous ESMAAtlas and AFDB light clusters contain between 444 and 714 proteins and are predominantly alpha-helical. Nine out of ten representative proteins are from ESMAAtlas and originate primarily from marine metagenomes. Functional predictions from DeepFRI show enrichment in biological process (BP) and cellular component (CC) ontology categories, including functions such as establishment of localization, membrane protein complex assembly, cell periphery organization, and broader cellular component organization. In contrast, the ten largest heterogeneous ESMAAtlas and AFDB dark clusters are smaller – ranging from 37 to 74 proteins – and exhibit far greater structural diversity, encompassing alpha, beta, and alpha-beta classes. All representative proteins in these clusters come from ESMAAtlas and originate from a mix of environments – host-associated, engineered, and freshwater metagenomes – and are associated with functions spanning all Gene Ontology categories, including nucleus, ribosome, cell periphery, plasma membrane, and DNA binding.

Page 11.

The conclusions around Panel C are affected by the size limitations of AFDB.

As far as we understand, relating to the comment re p. 5, Fig. 5C is correct - we use a cut off at 1300 residues, so no outliers from AFDB are considered. Structures longer than 1300 residues are presented in Fig. 5E since the statistics are very low and we decided to create a bar plot rather than a box plot. In Fig. 5C, we also included the mean \pm standard deviation for each bin to more clearly highlight the differences.

Page 14 and 15.

The conclusion section reads well, but there are no caveats with the work performed, or the effect of pLDDT, IDPs or issues with pLM-based structure predictors. Assuming that the snapshot of structure space is correct and not biased by the limitations of current methods (particularly ESMFold) can be misleading.

We thank the reviewer for this important suggestion. We appreciate the reviewer's point and agree that interpreting the structural landscape from pLM-based predictors would be misleading. That's why, in our study, we took a conservative approach by focusing on high-confidence ESMFold predictions (hclust30 subset of ESMAtlas) to mitigate quality concerns. However, we acknowledge key limitations: our evaluation of AlphaFold predictions depends on pLDDT, which may not fully reflect structural uncertainty, particularly in disordered regions. We did not explicitly address intrinsic disorder, as doing so would require MSA-based methods and considerable computational resources, placing it outside the scope of this work. Additionally, our filtering and clustering tools (e.g., Foldseek and Geometricus) have sensitivity limitations, and our reliance on 2D embeddings restricts the resolution of structural patterns. We added the following paragraph to the Discussion:

There are some limitations to the study and approach taken by us. Due to the volume of data, we were unable to comprehensively address intrinsic disorder in proteins (except approximating it using pLDDT), and phenomena which are associated with disordered proteins, such as fold-switching, moonlighting or preferred conformations [67–70]. Accurate characterization of disordered regions typically relies on MSA-based methods that are computationally intensive and beyond the scope of our current framework. Additionally, since we used non-redundant sequences as inputs, we are unable to directly examine the sequence-structure relationship, which could, for instance, offer valuable insights when exploring taxonomic diversity. Moreover, the tools used for filtering and clustering (such as Foldseek and Geometricus) introduce limitations in sensitivity, potentially missing finer-grained structural relationships. The use of 2D embeddings, while valuable for visualization, also constrains the complexity of structural patterns that may be captured. Nevertheless, within these methodological bounds, our analysis offers a meaningful and informative view of the protein structure space.

Supplementary.

The benchmarking of the parameters used for Foldseek clustered might need a section on single-domain chains, as variability in linker regions seems to be a limitation of full-chain clusters as seen in TED and the AFDB clustering paper in Nature. What's the RMSD within clusters? And is the TMscore cutoff used qtmscore, ttmscore or alntmscore? A normalised version of qtmscore/ttmscore is often more accurate than alntmscore. Since limiting the TMscore to the aligned region might cause issues if the alignment is poor or one of the two

proteins is a fragment. What was the flag used in Foldseek? Coverage of both query and target (bidirectional)?

We thank the reviewer for the insightful comments regarding the quality of Foldseek clustering. We agree that this approach has limitations, particularly when dealing with multi-chain proteins. For example, we acknowledge that using *qtm_score* or *tmscore* instead of *alntmscore* could improve benchmarking accuracy. However, for the sake of consistency with the methodology presented by Barrio-Hernandez et al. in *Nature* (“Clustering predicted structures at the scale of the known protein universe”), we chose to follow a similar procedure. This alignment was necessary to enable a direct comparison between ESMAtlas, MIP, and AFDB clusters. To enhance robustness, we independently tuned the e-value and coverage parameters for each database (MIP, hclust30, and the final combined set). We then performed a sanity check using US-align, which confirmed satisfactory TM-score distributions (see Figs. S4–S5, S11–S13, and S16). We used the default Foldseek coverage mode (`cov_mode = 0`), which considers the coverage of both query and target. This has been clarified in the Supplement, under the “*Structural clustering*” section, with the following sentence added to the first paragraph:

“In all cases, we used `cov_mode = 0` (coverage of query and target).”

Thanks for the great work and the pleasant read! I’m sure the community will benefit from this!

Thank you very much!

(Remarks on code availability)

The code is accessible and well documented. It includes code as python notebooks to recreate figures in the manuscript.

Reviewer #2 (Remarks to the Author)

In this study, the authors performed a large-scale structural clustering using protein structural models derived from three databases: AFDB, ESMAtlas, and MIP (Microbiome Immunity Project). Using the Foldseek program for clustering, they constructed a structural space that contains a single representative for each cluster derived from the individual databases. The principal structural conclusion of this extensive comparison is that the three databases exhibit structural complementarity, with AFDB and ESMAtlas displaying significant overlap within the structural space. In contrast, the contribution of MIP proteins to this space is more difficult to interpret, given that MIP predominantly comprises shorter and/or single-domain structures. Additionally, the authors observed that globular or ordered structures or structural motifs change gradually throughout the structural space, and they identified a specific region that likely contains intrinsically disordered proteins (IDPs), primarily represented by lower-quality AFDB models.

Subsequent functional annotation with deepFRI enabled the authors to define groups of structures associated with general functionalities—namely, “Cellular Processing and Signaling,” “Metabolism,” and “Information Storage and Processing.” They noted that proteins assigned to these groups tend to cluster in specific regions, a phenomenon the authors describe as “functional locality.” Additionally, an entire section is dedicated to the putative functional annotation of novel folds, most of which are derived from MIP examples.

Finally, with the obtained data, authors designed a web server to display the obtained information. The web server is both visually appealing and user-friendly.

While I acknowledge the considerable effort undertaken by the authors to execute such a massive analysis of structural similarities across proteins from three distinct databases, I remain concerned about the biological relevance of the conclusions drawn from this work. Specifically, the actual contribution of MIP proteins to the overall structural space delineated by AFDB and ESMAtlas is challenging to assess due to the size and organization of the proteins in MIP. Furthermore, considering that AFDB likely contains millions of homologous proteins that also appear in ESMAtlas, the conclusions regarding overlapping databases and their structural similarities might be inherently biased.

We agree that the MIP database differs from AFDB and ESMAtlas, particularly in terms of size. However, this difference supports our point that including or excluding MIP does not significantly alter the overall comparison between AFDB and ESMAtlas. In fact, the benefits of including MIP outweigh those of excluding it. For example, we can separately analyze the heterogeneity of clusters from both AFDB-ESMAtlas and AFDB-ESMAtlas-MIP (Fig. 5A-B) and identify novel folds among single-domain MIP structures (Fig. 4). Additionally, we can draw meaningful conclusions about the similarities and complementarities between AFDB and ESMAtlas (Fig. 1C, Fig. 5A), where MIP does not introduce bias. Regarding the overlap between AFDB and ESMAtlas, we addressed potential bias by removing structural redundancy within each database prior to comparison. We do not consider homology between databases to be a major source of bias, but rather the issue lies with protein IDs

that may appear in both databases. This, however, is difficult to evaluate due to the lack of available mapping between UniProt and MGNify IDs.

We have also elaborated on the biological significance of our work in the Introduction and Discussion sections. Please, see the answer to your final question (*"Overall, I consider the manuscript to be a noteworthy"*).

From a biological perspective, it may have been more informative to express the overlapping and differences between AFDB and ESMAtlas in terms of protein classes/taxonomies and also that these differences be available as tables in the web site/supplementary files. Furthermore, same information as taxonomic groups (e.g., eukaryotic, prokaryotic, archaeal, viral) and/or functional categories, could be used to characterize the heterogeneity within a given cluster or structural region more effectively than merely considering the origin of each structural model.

We thank the reviewer for this excellent suggestion. Recognizing the importance of taxonomy analysis, we have added a new figure to the manuscript (now Figure 6) along with a dedicated section in the main text titled *"Taxonomic imbalance in structure databases may cause a predictive bias"*. The methodology is described in detail in the corresponding subsection of the Methods (*"Taxonomy analysis"*). Additionally, we expanded the Supplement to include Figures S60–S62 and Tables S6–S7. Our main conclusions are: 1) predictions of prokaryotic proteins exhibit higher accuracy, mainly due to the lower proportion of disordered regions, 2) there is a non-random distribution of taxonomic groups across several structural clusters, which overall indicates 3) a taxonomic imbalance in current structural datasets. Unfortunately, differences in taxonomic assignment methods – AFDB and MIP IDs use NCBI taxonomy, while ESMAtlas proteins derive from metagenomic studies and are assigned to biomes – prevent a direct comparison across all databases. We have added the following to the Discussion:

Due to disparity in how the two main structural databases have been constructed, direct taxonomic comparison within the protein space is difficult. It is apparent, though, that the majority of protein structures currently available represent prokaryotic organisms, predominantly bacteria.

Nevertheless, to the best of our knowledge, no previous study has analyzed ESMAtlas taxonomy stratification or compared it with AFDB, making this contribution a valuable extension of our work's impact.

The exploration of the structural space of globular domains associated with intrinsically disordered regions (IDRs) is an intriguing aspect of the manuscript; however, relying solely on average pLLDT values to identify IDRs or IDPs could be problematic and should ideally be complemented with additional disorder estimation methods.

We do agree with the reviewer that intrinsic disorder is an important phenomenon which should not be ignored in the context of the protein universe. Our dataset is based on ~1.5 M sequences, so it is a sizable challenge in itself. The most efficient methods for IDR/IDP detection rely on multiple sequence alignments, which are not feasible for us to generate at the scale that it would require. Otherwise, looking at single-sequence predictors, we see

evidence that they are no better than using AlphaFold pLDDT (currently references 61-64 in the updated manuscript). In fact, there is evidence that AlphaFolds pLDDT scores are currently one of the best methods. Wilson and co-workers (ref. 64 in the manuscript), for example, compared AlphaFold pLDDT scores to traditional methods of predicting IDPs and IDRs on the DisProt and DisProt-PDB datasets. They found that pLDDT-based methods performed as other methods and were the best performing on the DisProt-PDB dataset. We however acknowledge that AlphaFolds pLDDT scores are context dependent, and that low pLDDT may not indicate intrinsic disorder but simply disorder in which the model was generated. We consistently emphasize throughout the paper that low pLDDT scores may represent a broader set that encompasses, but is not limited to, intrinsically disordered proteins. We have added additional references and a comment to the Discussion section:

However, considering that many of them are likely disordered (see [61–64] for correlation between AlphaFold pLDDT and disorder), this highlights the need for improved functional annotation of intrinsically disordered proteins (...)

Regarding the functional analysis and the notion of “functional locality” described by the authors, one must consider whether this observation is simply a consequence of homologous proteins naturally occupying similar regions within the structural space. In this context, the pertinent null hypothesis is: What additional information does structural data provide in revealing similarities between proteins? In other words, would clustering all these proteins based solely on sequence data yield comparable results? Although it is well understood that protein structures are generally more conserved than their sequences—thereby potentially improving the detection of remote homologues or similarities—it should be noted that, while Foldseek offers enhanced speed, it is approximately 15% less sensitive than methods such as Dali (Kempen et al., 2023). Could the authors please clarify how they believe that the reduced sensitivity of their method affects the overall findings, particularly in the section devoted to new folds? Additionally, is there any supplementary functional or biological information available that could further support the predicted functions of the examples presented in Figure 4?

Regarding the first part, this is a very intriguing question. Protein structures generally evolve more slowly than amino acid sequences, and the relationships among sequence similarity, structural similarity, and functional similarity are complex and not strictly linear. While sequence clustering captures many functional relationships, it can miss important functional connections that are only recognizable through structural analysis, especially for distantly related proteins. One intriguing aspect that we do not explore in our study but would be valuable, is to estimate sequence similarity for each input in our dataset ((which consists of non-redundant sequences; see Fig. 1A) and, for example, analyze taxonomic diversity. This is, however, a massive challenge due to the size of the input databases and falls beyond the scope of our study. We added the following to the discussion:

Additionally, since we used non-redundant sequences as inputs, we are unable to directly examine the sequence-structure relationship, which could, for instance, offer valuable insights when exploring taxonomic diversity.

As for the second part, we considered using Dali/TMalign, but it was unfeasible for the scale of our study. Instead, we conducted a comprehensive benchmark comparing Foldseek and

US-align (a more robust version of TM-align) – see Figs. S4–S5, S11–S13, and S16. Additionally, we fine-tuned Foldseek parameters for each database to minimize bias. Regarding novel folds, these were identified based on an all-vs-all TM-align comparison against PDB in Koehler Leman et al.'s “Sequence-structure-function relationships in the microbial protein universe” paper published in Nature Communications, so their detection is not affected by Foldseek's sensitivity limitations. We have added the appropriate reference in the sentence “*MIP novel folds as a subset of the entire MIP dataset [25]*” in the second paragraph of section “*High-level functional categories are localized in specific regions of the structure space*”.

Regarding the additional information that could support the functional predictions, there are other tools e.g. TMHMM to predict transmembrane regions [Krogh *et al.*, 2001], which can be used to cross-validate the annotations. Also knowledge-based information, such as the habitat of the host organism can also verify whether the predicted functions are likely to be present. We have added the following at the end of first paragraph in section “*Exploring functional diversity in novel protein folds*”:

The output of annotation from all core ontologies (BP, CC and MF), generated by deepFRI agree with the predicted function, by e.g. determining its subcellular location, which can be cross-validated e.g. with TMHMM [47]. Moreover, applying knowledge-based information, such as the habitat of the host organism, i.e. to confirm whether this could be a methanogen, as in the case of Fig. 4A or displaying host-pathogen interactions, as in the case of Fig 4C, helped us verify whether the predicted functions are likely to be present.

Regarding the web server, while it is well designed and visually appealing, I encountered difficulties when attempting to locate the examples mentioned in Figure 4, which would help in understanding the uniqueness of those folds. Moreover, I attempted to download the data from the MIP database; however, it appears that the entire dataset must be downloaded in order to access just those specific examples. Related to the search capacity, Would it be possible to search for any protein contained in a given cluster besides the one being the representative one?

Those are excellent suggestions, thank you! We have significantly enhanced the web server, which now supports downloading PDB files, searching by all IDs (not just representatives), searching by functional terms (GO-term names), filtering by taxonomy, displaying functional annotations from deepFRI as well as origin, redirecting to data repository (AFDB, ESMAtlas and NCBI web pages), and viewing all proteins within a given cluster - see updated Fig. 7.

Overall, I consider the manuscript to be a noteworthy piece of work, but I believe it would benefit from addressing the concerns raised above and further elaborating on the biological implications of their findings.

Thank you very much for your valuable comments. We believe that our updated version effectively addresses the reviewer's concerns. We have also made additional efforts to address the biological significance of our work in the *Introduction* and *Discussion* sections.

In Introduction we have added:

Specifically, what's missing is a shared structural and functional space – a unified reference frame that allows proteins from disparate datasets to be compared meaningfully. Such a framework would support cross-dataset biological inference, enabling systematic comparisons across structure predictors, taxonomic groups, and sequence contexts. It would also help in identifying complementarity and redundancy, revealing which datasets contribute novel folds and which primarily reinforce known structural motifs. By incorporating structure-based function annotations into this space, we can identify where proteins with similar biological roles cluster together, even in the absence of sequence similarity. This makes this space a powerful discovery tool – one that facilitates the exploration of uncharacterized folds, improves functional annotation pipelines, and helps prioritize proteins of interest in applications ranging from metagenomics to protein engineering. Further, this reference frame is extensible, offering a mechanism to incorporate future datasets for direct structural and functional comparison.

In the Discussion section we have added:

This study attempts to grapple with a new reality in which high-quality protein 3D structure models have become a commodity. In 2021-2022 with the releases of AlphaFold Database (AFDB), ESMAtlas, and other resources, the research community went from a reality in which there were ~200,000s of high-quality protein structures available, to a reality in which around 1 billion models are available (10,000x change). The field had decades to learn, adapt and develop a deep understanding and best working practices to be dealing with protein sequences at a scale. The rapid technological shift and abundance of 3D protein structure data occurred left a conceptual gap in how we deal with, contextualize, or learn from this newly available perspective on biology. Our study offers an outlook on how this can be achieved and proves that in fact it is possible to draw biologically meaningful conclusions from a protein structure landscape perspective.

(Remarks on code availability)

Reviewer #3 (Remarks to the Author)

In this study, the authors provide a study of the structural and functional landscape of the protein universe. They collate a large number of predicted structures from AFDB, ESM Atlas, and MIP. They also functionally annotate the proteins using Deepfri, a computational function prediction method. The authors provide results on a web server, which is searchable by UniProtID or function. Overall, there was a large investment in this body of work, but it left me wondering about the utility and novelty of this study. Furthermore, I have identified certain methodological issues.

This work relies primarily on predicted structures and functions. It is not clear if there are any experimentally produced structures (e.g. via X-ray crystallography or NMR) nor experimentally annotated functions. Those can help provide verified structure or function data to what is otherwise a compilation of structural and functional predictions.

We thank the reviewer for this comment. We have not used experimental structures or experimental annotations of functions, except CATH representatives in Fig. 2B used for illustrative purposes. To clarify this, we replaced the first sentence in the third paragraph of Introduction from “*In this study, we analyze three large protein structure datasets originating from distinct sources.*” into “*In this study, we analyze three large protein structure model datasets originating from distinct sources.*”, as well as extended the first sentence in the first paragraph of section “*High-level functional categories are localized in specific regions of the structure space*” with “*We performed functional analysis using deepFRI (...) which predicts Gene Ontology terms along with corresponding confidence scores.*”.

Webserver: since the proteins used have been clustered, it means that only a representative of a protein group will be represented in the server. While that is a sound strategy for creating a structural landscape, it makes it impossible to search for a protein using any uniprot ID, since the user needs to know, for example, what is the representative uniprotID chosen for TP53? (I could not find it).

Yes, we agree that this was inconvenient. We have improved it in the current version of the web server (see updated Fig. 7). Now, when searching for a specific cluster member, the cluster representative point is displayed on the landscape (since mappings exist only for representatives), while the structure and function predictions are shown for the queried protein.

While the authors provide a measure of confidence in structure prediction using PLDDT they do not provide the equivalent function prediction for the DeepFri model, which is a confidence level of 0-1. Furthermore, PLDDT is a per-residue metric, not a per-protein metric. It is not clear that a protein with an overall low PLddT does not have a reliable structure, it may mean that it just contains many disordered regions, but the non-disordered regions are predicted correctly. Therefore, it is not clear what purpose the PLDDT filter serves.

We provide deepFRI scores for deepFRI, e.g. Fig. 4, and generally apply a default filter of score ≥ 0.3 for deepFRI predictions, following Gligorijevic et. al (2021) Nat Comms and Koehler Leman et al. (2023) Nat Comms. Regarding the web server, we list function predictions for a selected protein for lower scores (≥ 0.2) to allow the user to decide which functions might be relevant. Filtering by pLDDT in the manuscript allowed us to draw some qualitative conclusions on where globular and presumably disordered proteins (or proteins that pose problems to AlphaFold2) may be placed in the structure space. While we agree that pLDDT is a local quality measure, a mean pLDDT has become a de facto standard metric used as a proxy for overall prediction quality. Most importantly, unlike GDT-TS or TM-score values (or pTM values), pLDDT values are well-calibrated and readily available for each structure in our space. We believe that experienced users of the web server can use this functionality for their own purposes. Additionally, in the first paragraph of section “*Web server allows for an interactive exploration of the protein universe*” we rephrased the sentence “*AFDB pLDDT, which is particularly helpful for identifying disordered proteins*” to “*AFDB pLDDT, which may be particularly helpful for identifying disordered proteins*” to reflect a more cautious interpretation.

The webserver provides a static map of the dimensionality reduction of this study. It is zoomable, but not translatable. Furthermore, I would expect that clicking on any of the dots (the clustering representative) would provide me with information on this particular protein, but that does not happen. Therefore, it is unclear why this mapping is provided in a zoomable/clickable format. (Unless there is a bug, and this function is indeed there, as it seems to be intuitive that it should be).

We sincerely apologize for the inconvenience. The reviewer likely encountered this issue during our maintenance period while we were testing new utilities. The web server is now fully operational, stable, and updated with new features.

The dimensionality reduction map purports to provide a variety of protein structure and function space. But without the ability to rotate, translate, and color-code the map by functions or structure.

The dimensionality reduction maps shown in Fig. 1B-C and 3B can be reproduced using the code available in the project's GitHub repository (<https://github.com/Tomasz-Lab/protein-structure-landscape>), specifically the notebook `Figure_1BC_and_3B.ipynb` and `data_loader.ipynb`. Thanks to that, the user can customize the maps according to their needs. Additionally, the notebooks `embeddings.ipynb` and `dimensionality_reduction.ipynb` can be used to create embeddings on external datasets, and transform them using our dimensionality reduction approach. The web server already offers the ability to translate and zoom within the space. While custom color-coding is not currently supported, different databases are distinctly color-coded by default. Users can also filter by categorical features (SuperCOGs, taxonomy, or database) or continuous features (AFDB pLDDT, length) to focus on specific aspects of the landscape.

Finally, I do not understand the novelty of this study, and the value it provides to advancing our understanding of protein structure and function landscape. The authors do not articulate that anywhere, and the only justification I can find was in the abstract and the discussion that

this study “lays the groundwork” for future studies. However, it is not said how this study lays the groundwork for future studies.

To our knowledge, no one has previously compared AFDB and ESMAtlas in terms of their structural and functional aspects. We have identified structural complementarity and functional locality, both of which are, in our opinion, significant findings. We also added taxonomy information (see “*Taxonomic imbalance in structure databases may cause a predictive bias*” section in the Results) and compared the databases in this context. We elaborated on the biological significance of our work in the Introduction, Discussion and in the revised abstract to more accurately reflect our findings.

In Introduction we have added:

Specifically, what’s missing is a shared structural and functional space – a unified reference frame that allows proteins from disparate datasets to be compared meaningfully. Such a framework would support cross-dataset biological inference, enabling systematic comparisons across structure predictors, taxonomic groups, and sequence contexts. It would also help in identifying complementarity and redundancy, revealing which datasets contribute novel folds and which primarily reinforce known structural motifs. By incorporating structure-based function annotations into this space, we can identify where proteins with similar biological roles cluster together, even in the absence of sequence similarity. This makes this space a powerful discovery tool – one that facilitates the exploration of uncharacterized folds, improves functional annotation pipelines, and helps prioritize proteins of interest in applications ranging from metagenomics to protein engineering. Further, this reference frame is extensible, offering a mechanism to incorporate future datasets for direct structural and functional comparison.

In the Discussion section we have added:

This study attempts to grapple with a new reality in which high-quality protein 3D structure models have become a commodity. In 2021-2022 with the releases of AlphaFold Database (AFDB), ESMAtlas, and other resources, the research community went from a reality in which there were ~200,000s of high-quality protein structures available, to a reality in which around 1 billion models are available (10,000x change). The field had decades to learn, adapt and develop a deep understanding and best working practices to be dealing with protein sequences at a scale. The rapid technological shift and abundance of 3D protein structure data occurred left a conceptual gap in how we deal with, contextualize, or learn from this newly available perspective on biology. Our study offers an outlook on how this can be achieved and proves that in fact it is possible to draw biologically meaningful conclusions from a protein structure landscape perspective.

Minor: The data could use better documentation, e.g. how to open the parquet files provided and a readme file detailing their format

Yes, absolutely. We have improved the README in the project’s GitHub repository to better describe its contents. In particular, the parquet files in the data folder can be opened using the `data_loader.ipynb` notebook in the notebooks directory (with the `pandas.read_parquet()`

Python function). The parquet format is more portable and disk-space efficient compared to text formats like CSV or TSV.

P. 14 "...gene-coding proteins...". genes code for proteins, not vice-versa.

We replaced "*gene-coding proteins*" with "*genes coding for proteins*".

Reviewer #3 (Remarks on code availability):

The code is not intended to be replicable on another machine, as far as I can tell.

The codebase is designed to be reproducible: all analyses presented in the paper can be replicated using our GitHub repository: <https://github.com/Tomasz-Lab/protein-structure-landscape> and Figshare data repository https://figshare.com/articles/dataset/Large_protein_databases_reveal_structural_complementarity_and_functional_locality/27203073, where, additionally, deepFRI v1.1 model weights, and training data can be found as well as embeddings needed to create the 2D mappings for protGPT2 and BFVD datasets. We have enhanced the README for clarity and usability, and added supplementary data files (e.g., taxonomic information) and notebooks – including tools for applying our dimensionality reduction pipeline to other datasets. Additionally, the web server has been significantly updated to provide an improved user experience when exploring our dataset.